# Understanding the Gains from Repeated Self-Distillation

**Divyansh Pareek**     **Simon S. Du**     **Sewoong Oh**
Paul G. Allen School of Computer Science and Engineering
University of Washington, Seattle, WA
{dpareek,ssdu,sewoong}@cs.washington.edu

## Abstract

Self-Distillation is a special type of knowledge distillation where the student model has the same architecture as the teacher model. Despite using the same architecture and the same training data, self-distillation has been empirically observed to improve performance, especially when applied repeatedly. For such a process, there is a fundamental question of interest: How much gain is possible by applying multiple steps of self-distillation? To investigate this relative gain, we propose studying the simple but canonical task of linear regression. Our analysis shows that the excess risk achieved by multi-step self-distillation can significantly improve upon a single step of self-distillation, reducing the excess risk by a factor as large as $d$, where $d$ is the input dimension. Empirical results on regression tasks from the UCI repository show a reduction in the learnt model's risk (MSE) by up to $47\%$.

## 1 Introduction

Knowledge distillation [12] was initially proposed as a way to transfer the knowledge learnt by a larger teacher model to a smaller student model, which can then be deployed in limited resource settings. The process is as follows: Train a teacher ($T$) model using ground-truth labels, then use its predictions to supervise the training of a student ($S$) model via a combined per-sample loss,

$$\xi \cdot \boldsymbol{\ell}\big(\hat{y}_T, y_S(\theta)\big) + (1 - \xi) \cdot \boldsymbol{\ell}\big(y, y_S(\theta)\big) \,, \tag{1}$$

where $\boldsymbol{\ell}$ denotes the loss function, $y$ is the ground-truth label, $\hat{y}_T$ denotes the teacher's prediction, and $y_S(\theta)$ denotes the student's prediction, parameterized by the learnable $\theta$. The extra hyperparameter $\xi$ is called the imitation parameter [25], generally restricted to $\xi \in [0, 1]$. It gives additional freedom to the student to balance importance between labels and teacher's predictions. The student trained via this distillation objective (i.e., utilizing the teacher's predictions through $\xi \neq 0$) has been widely observed to generalize better than when trained only on the labels (i.e., $\xi = 0$). This gain has been attributed to 'dark knowledge' that is $(i)$ impossible to be directly extracted from the training data by the small model, but $(ii)$ easily learnt by the large model and transferred to the small model.

Challenging this interpretation, Li et al. [20] and Furlanello et al. [10] empirically observed performance gains through distillation even when the teacher and student are same-sized models. One can set $T$ and $S$ to have the *same architecture*, and $S$ trained with the objective in Eq. (1) outperforms $T$. This is referred to as Born-Again Networks (BANs) or *Self-Distillation* (SD). Furthermore, repeatedly applying self-distillation on the same training data with a student model having the same architecture provides additional gains on benchmark datasets and architectures [10, 36, 44]. At each step, the student from the previous step acts as the teacher used to train a new student model under the self-distillation loss of Eq. (1). For such *multi-step self-distillation*, there is a fundamental question of interest: *How much more gain can we get by repeatedly applying self-distillation?*

Recently, Das and Sanghavi [9] provided theoretical understanding of the original one-step self-distillation. For the canonical task of fixed design linear regression, considering the standard ridge

38th Conference on Neural Information Processing Systems (NeurIPS 2024).

estimator as both the teacher and student model, [9] showed that there is indeed a regime of problem instances in which the *optimal* student (i.e., with optimally tuned ridge parameter $\lambda$ and imitation parameter $\xi$) can provably achieve a strictly lower test error than the *optimal* teacher (i.e. with optimally tuned $\lambda$). However, the amount of this gain has not been characterized in closed form, and can only be numerically evaluated for a given problem instance. Inspired by this work, we aim to study the performance gains from multi-step self-distillation under linear regression.

**Contributions.** We summarize our contributions below.

- Under the fixed design linear regression defined in Section 3.1, we show that the *optimal* multi-step self-distilled model (i.e., each $\xi$ value at each step is optimized for the validation accuracy of the final multi-step self-distilled model) can achieve a test error that is a factor of $d$ smaller than the *optimal* one-step self-distillation (Theorem 1), under certain assumptions on the problem parameters (Assumption 2). Here, $d$ is the dimension of the input. Our analysis in Theorem 1 suggests that the sequence of $\xi$ parameters provides additional freedom that can control the spectrum of eigenvalues of the linear estimator. Optimally choosing these $\xi$ parameters can significantly reduce the variance of the estimator, leading to a factor of (up to) $d$ difference in the overall test errors of multi-step SD compared to 1-step SD. We note that Das and Sanghavi [9] also observed a bias-variance tradeoff associated with the $\xi$ parameter for 1-step SD compared to the ridge, which was the reason behind 1-step SD strictly outperforming the ridge.

- We demonstrate the necessity of the main assumption (Assumption 2) both theoretically (Theorems 2 and 3) and numerically (Figure 3). Further, we provide a lower bound for the test error that any repeated SD can achieve, and show that only $r$ steps of SD (with optimally chosen $\xi$ at each step) are sufficient to achieve this, when the input data matrix has rank $r$ (Theorem 4).

- By capturing the functional form of the test error in $\xi$ (Theorem 5), we also show a method to practically select the $\xi$ parameters for real-world regression tasks. In Section 5, we empirically show that this theoretical insight leads to selecting effective $\xi$ values, which can indeed achieve a lower test error on real-world regression tasks.

## 2  Related Work

**Knowledge distillation and self-distillation.** Hinton et al. [12], Ba and Caruana [3] proposed knowledge distillation to transfer knowledge learnt by large teacher models into smaller student models without any substantial performance drop (e.g., [30, 31, 14, 8, 33, 24, 32] and surveys in [11, 13]). Distillation also provides interpretability [23], robustness to adversarial examples [28], and defense against backdoor attacks [39, 21, 35], although stronger backdoor attacks have been proposed that bypass distillation defense [17]. Perhaps surprisingly, empirical observations show that performance *improves* when a teacher model is distilled into a student model with the same architecture on the same training data (*self-distillation*). Performance gains with one-step self-distillation of the form Eq. (1) were first demonstrated by Li et al. [20] for AlexNet on YFCC100M. Further gains can be achieved by repeating self-distillation, as shown for the DenseNet architecture family on CIFAR10 and CIFAR100 [10, Table 2]. To empirically explain such gains, Zhang and Sabuncu [44] measured prediction uncertainty on the same multi-step experiments and offered an interpretation that soft labels capture sample-level uncertainties. Yang et al. [36] also reproduced the same experiments and explained the gains as knowledge refinement on the class similarities. We will analytically study the gains that can be achieved with such repeated self-distillation.

Many variations of self-distillation have also been proposed. Snapshot Distillation [37] tries to treat previous checkpoints (snapshots) of the same model as the teacher. Zhang et al. [43] employ a group of collaborative students with no teacher. Zhang et al. [42, 41] use it for model self-improvement, and DINO [7] adopts self-distillation for self-supervised learning. Knowledge distillation is also popular for transfer learning, where the student model is trained on a different dataset than the teacher model [38, 40, 1], which is not a setting we address. With the recent scaling of data, [45], [22] are relevant works using a teacher model for either label editing or data reweighing.

**Theory of distillation and self-distillation.** Theoretical understanding of distillation started with Phuong and Lampert [29] studying linear student networks. Mobahi et al. [27] studied self-distillation theoretically in the restricted setting of $\xi = 1$ (i.e. only teacher supervision, no ground-truth labels), showing that in this setting, the SD process acts as a regularizer, with a few steps of SD helping, but further steps hurting model performance. We study the setting where $\xi$ is not restricted to 1,

and show a different conclusion. In particular, we observe that more steps of SD always provide an improvement, if the $\xi$ parameters are chosen optimally. Allen-Zhu and Li [2] analyzed a stylized setting, where a different view of the data is learned by different models, and show how ensemble methods can combine the views, achieving improved test accuracy. This framework is used to show how self-distillation can also improve model accuracy by implicitly performing ensembling. Menon et al. [26] theoretically studied distillation in the classification setting, and also observed a bias-variance tradeoff underlying teacher supervision. Das and Sanghavi [9] theoretically studied one-step self-distillation for fixed design linear regression and binary classification, and showed that the student can provably achieve a lower test error than the teacher. We take inspiration from them and study the multi-step SD to characterize this performance gain, showing that the multi-step SD can outperform one-step SD by a large factor. Borup and Andersen [4] also studied multi-step SD and obtained an analytical form for the $k$-step SD similar to ours [4, Theorem 4.1]. The crucial difference is the freedom of the $\xi$ parameters being different at each step of self-distillation. Whereas [4, Lemma 4.2 and Theorem 4.3] assume the $\xi$ values at each step are equal (similar to [27]), concluding that after a point, more steps of SD will result in a poorer performing model (similar to [27]); our main result (Theorem 1) is different as it says that subsequent steps of SD strictly provide more freedom, and that the best multi-step SD can outperform the best 1-step SD by an $\Omega(r)$ factor. Similar to us, Jeong and Chung [16] also take inspiration from [9], [16] however aim to provide understanding of multi-step self-distillation in the multi-class classification setting.

## 3 Problem formulation and background on self-distillation

Focusing on the simple but canonical task of linear regression, we investigate the performance gain from applying repeated self-distillation.

### 3.1 Linear regression

For the observed response $Y \in \mathbb{R}$ and the covariate $X \in \mathbb{R}^d$, the following assumption is standard in linear regression, e.g., [9].

**Assumption 1.** *There exist $\theta^\star \in \mathbb{R}^d$ and $\gamma > 0$ such that (i) $\mathbb{E}[Y|X] = \langle \theta^\star, X \rangle$; (ii) $\mathrm{Var}[Y|X] = \gamma^2$ for all $X \in \mathbb{R}^d$; and (iii) $(Y - \mathbb{E}[Y|X]) \perp\!\!\!\perp X$, i.e. the label noise is independent of $X$.*

The training set of size $n$ is denoted by $\mathbf{X} \in \mathbb{R}^{d \times n}$, the collection of covariates, and $\mathbf{Y} \in \mathbb{R}^n$, the responses; $\mathbf{Y} = \mathbf{X}^\top \theta^\star + \boldsymbol{\eta}$, with $\boldsymbol{\eta}$ satisfying $\mathbb{E}[\boldsymbol{\eta}] = 0$, $\mathbb{E}[\boldsymbol{\eta}\boldsymbol{\eta}^\top] = \gamma^2 \mathbf{I}_n$. The problem instance is defined by its parameters $(\mathbf{X}, \theta^\star, \gamma^2)$, treating $\mathbf{X} = [X_1, X_2, \cdots X_n]$ as fixed but $\mathbf{Y}$ as random. The training set $(\mathbf{X}, \mathbf{Y})$ is one occurrence of the random noise $\boldsymbol{\eta} \in \mathbb{R}^n$. In this *fixed design* setup, the excess risk of an estimator $\hat{\theta}$ is defined using the standard $\hat{\Sigma}_n$-norm, $\|v\|_{\hat{\Sigma}_n} = \|\hat{\Sigma}_n^{1/2} v\|_2$, as

$$\mathrm{ExcessRisk}(\hat{\theta}) \quad := \quad \mathbb{E}_{\boldsymbol{\eta}} \left[ \|\hat{\theta} - \theta^\star\|_{\hat{\Sigma}_n}^2 \right] , \tag{2}$$

where $\hat{\Sigma}_n := (1/n)\mathbf{X}\mathbf{X}^\top$ is the covariance matrix, and the expectation is over the randomness in $\boldsymbol{\eta}$. Measuring the error in the $\hat{\Sigma}_n$-norm ensures that the signal-to-noise ratio is uniform in all directions. The popular ridge estimator serves as a baseline, using a single hyperparameter $\lambda > 0$:

$$\hat{\theta}(\lambda) \quad := \quad \arg\min_{\theta \in \mathbb{R}^d} \left( \|\mathbf{Y} - \mathbf{X}^\top \theta\|^2 + \lambda \|\theta\|^2 \right) \quad = \quad \left( \mathbf{X}\mathbf{X}^\top + \lambda \mathbf{I}_d \right)^{-1} \mathbf{X}\mathbf{Y} . \tag{3}$$

We use $\boldsymbol{\Omega}_\lambda := \mathbf{X}\mathbf{X}^\top + \lambda \mathbf{I}_d$ throughout. We consider only $\lambda > 0$, but surprisingly, Kobak et al. [19] showed that the optimal penalty $\lambda^\star$ (one with the lowest risk) can indeed be negative. However we will largely work in the non-overparameterized case ($n > d$), where $\lambda^\star > 0$ holds.

### 3.2 Self-distillation

Applying the self-distillation loss in Eq. (1) to linear regression with hyperparameters $\lambda$ and $\xi$,

$$\hat{\theta}(\lambda, \xi) \quad := \quad \arg\min_{\theta \in \mathbb{R}^d} \left( \xi \|\mathbf{X}^\top \hat{\theta}(\lambda) - \mathbf{X}^\top \theta\|^2 + (1 - \xi) \|\mathbf{Y} - \mathbf{X}^\top \theta\|^2 + \lambda \|\theta\|^2 \right) \tag{4}$$

$$= \quad \left( \mathbf{X}\mathbf{X}^\top + \lambda \mathbf{I}_d \right)^{-1} \mathbf{X} \underbrace{\left( \xi \cdot \mathbf{X}^\top \hat{\theta}(\lambda) + (1 - \xi) \cdot \mathbf{Y} \right)}_{\text{New label}} \tag{5}$$

$$= \quad \underbrace{\left\{ (1 - \xi) \cdot \mathbf{I}_d + \xi \cdot \boldsymbol{\Omega}_\lambda^{-1} \mathbf{X}\mathbf{X}^\top \right\}}_{\text{Pre-conditioner: function of } (\lambda, \xi)} \cdot \underbrace{\boldsymbol{\Omega}_\lambda^{-1} \mathbf{X}\mathbf{Y}}_{\text{Ridge } \hat{\theta}(\lambda)} , \tag{6}$$

where $\xi \in \mathbb{R}$ is not restricted to the conventional $[0, 1]$ interval. This additional freedom is meaningful since it can result in a strictly better solution, as noted by Das and Sanghavi [9] both theoretically (Remark 3.6) and empirically (Table 1). It is worth noting that the optimization problem in eq. (4) remains convex for any $\xi \in \mathbb{R}$, since its hessian evaluates to $\xi \cdot 2\mathbf{X}\mathbf{X}^\top + (1-\xi) \cdot 2\mathbf{X}\mathbf{X}^\top = 2\mathbf{X}\mathbf{X}^\top \succeq 0$. On the other hand, the teacher and student use the same ridge penalty $\lambda$ for simplicity.

We call this estimator 1-*step self-distillation*. This can be interpreted as $(i)$ assigning new labels that combine the ground-truth labels with the teacher's predictions, or $(ii)$ pre-multiplying the usual ridge estimator with a pre-conditioner. Note that $\xi = 0$ recovers ridge. Das and Sanghavi [9, Theorem 3.8] show that under a certain condition, 1-step self-distillation strictly dominates ridge, i.e.,

$$\min_{\lambda \geq 0, \xi \in \mathbb{R}} \mathbb{E}_{\boldsymbol{\eta}} \left[ \|\hat{\theta}(\lambda, \xi) - \theta^\star\|_2^2 \right] \quad < \quad \min_{\lambda \geq 0} \mathbb{E}_{\boldsymbol{\eta}} \left[ \|\hat{\theta}(\lambda) - \theta^\star\|_2^2 \right] , \qquad (7)$$

where the risk is measured in the non-standard Euclidean norm. The same strict inequality can be shown under the standard $\hat{\Sigma}_n$-norm under a slightly modified condition stated in Proposition B.1. This naturally leads to a fundamental question: *How much more gain can we get by repeatedly applying self-distillation?*

Figure 1: The standard 1-step self-distillation defined in Eq. (1) with parameter $\xi$ and $k$-step self-distillation that repeatedly applies Eq. (1) with parameter $\xi^{(k)} = [\xi_1^{(k)}, \xi_2^{(k)}, \dots, \xi_k^{(k)}] \in \mathbb{R}^k$.

## 3.3   Repeated self-distillation

The standard repeated application of self-distillation starts with the teacher model, $T$ (which we also refer to as the zeroth model, $S_0$), and applies self-distillation sequentially for $k$ steps. At each step $i$, Eq. (1) is applied with the $(i-1)^{th}$ model, $S_{i-1}$ as the teacher, and the $i^{th}$ model, $S_i$ as the student, with an imitation parameter $\xi_i^{(k)}$, i.e., $\hat{\theta} \in \arg\min_\theta \left\{ \xi_i^{(k)} \boldsymbol{\ell}(\hat{y}_{S_{i-1}}, y_{S_i}(\theta)) + (1 - \xi_i^{(k)}) \boldsymbol{\ell}(y, y_{S_i}(\theta)) \right\}$ for $i \in [k]$. The collection of parameters is denoted by $\xi^{(k)} = [\xi_1^{(k)}, \xi_2^{(k)}, \dots, \xi_k^{(k)}] \in \mathbb{R}^k$.

This repeated self-distillation has been studied, for example, theoretically in [27] and empirically in [10]. We aim to understand its gain under linear regression, where we prove that

$$\hat{\theta}(\lambda, \underbrace{\xi^{(k)}}_{\in \mathbb{R}^k}) \;=\; \underbrace{\left\{ \left( 1 - \sum_{i=1}^{k} \bar{\xi}_i^{(k)} \right) \mathbf{I}_d + \sum_{i=1}^{k} \bar{\xi}_i^{(k)} \left( \boldsymbol{\Omega}_\lambda^{-1} \mathbf{X}\mathbf{X}^\top \right)^i \right\}}_{\text{Pre-conditioner: } \mathbf{P}(\lambda, \xi^{(k)})} \cdot \underbrace{\boldsymbol{\Omega}_\lambda^{-1} \mathbf{X}\mathbf{Y}}_{\text{Ridge } \hat{\theta}(\lambda)} , \qquad (8)$$

with $\bar{\xi}_i^{(k)} := (1 - \xi_{k-i}^{(k)}) \prod_{l=k-i+1}^{k} \xi_l^{(k)}$ for each $i \in [k]$, and we let $\xi_0^{(k)} = 0$. The proof that repeated SD with $\xi^{(k)} \in \mathbb{R}^k$ results in Eq. (8) is provided in Appendix C.2. Here $\xi^{(k)} \in \mathbb{R}^k$ denote the imitation parameters, $\lambda$ denotes the ridge coefficient for all the models, and $\bar{\xi}^{(k)} \in \mathbb{R}^k$ is a *reparametrization* of $\xi^{(k)} \in \mathbb{R}^k$ (details in Appendix C.2). We call this $k$-*step self-distillation*. Note the increasing flexibility in the pre-conditioner matrix. The increasing powers of $\boldsymbol{\Omega}_\lambda^{-1} \mathbf{X}\mathbf{X}^\top$ in the above expression are still numerically stable, since, for $\lambda > 0$, $\boldsymbol{\Omega}_\lambda^{-1} \mathbf{X}\mathbf{X}^\top$ is PSD with all eigenvalues in $[0, 1]$. As an aside, one can also consider a version of SD where the $i^{th}$ model receives supervision from all $S_{<i}$ instead of just $S_{i-1}$. Appendix C.1 shows that this version provides no extra representational capacity over the repeated version presented above, when all $k$ entries of $\xi^{(k)}$ are optimized as free parameters. Hence, the procedure in Figure 1 suffices for analysis.

## 4   Main results for linear regression

The main object of our study is to theoretically demonstrate the gains from repeated self-distillation. Concretely, we aim to show that there can be a significant multiplicative separation between the

excess risk achieved by $r$-step SD (Self-Distillation), where $r$ is the rank of the input $\mathbf{X}$; compared to the ridge estimator, as well as the 1-step SD (Section 4.1). The necessity of the two main assumptions is shown in Section 4.2. The sufficiency of $r$ steps of SD is shown in Section 4.3. In Section 4.4, we provide an exact characterization of the excess risk achieved by $k$-step SD (for any $k$).

## 4.1 The $r$-step self-distillation significantly improves upon the 1-step self-distillation

We show the desired separation under the following assumption (and two more mild technical assumptions specified fully in Appendix E).

**Assumption 2.** *Assume the following two conditions hold on the problem instance $\left(\mathbf{X}, \theta^\star, \gamma^2\right)$ :*

1. *No two non-zero singular values of $\mathbf{X}$ collide, i.e. $s_1 > s_2 > \cdots > s_r > 0$, where $\{s_j\}_{j=1}^r$ denote the non-zero singular values of the input data matrix $\mathbf{X}$ whose rank is denoted by $r$.*

2. *For a $\beta \in [0,1)$, there exists an index $j \in [r]$ such that $\langle \theta^\star, \mathbf{u}_j \rangle^2 \geq (1-\beta) \cdot \|\theta^\star\|^2$; where $\{\mathbf{u}_j\}_{j=1}^d$ denote the eigenvectors of $\mathbf{X}\mathbf{X}^\top$, $\mathbf{u}_1$ being the leading one.*

Assumption 2 is needed to show that $r$-step SD achieves a small excess risk in Eq. (9). The quantity $\beta$ captures the similarity of the (unknown) $\theta^\star$ to the eigenbasis directions. The case of $\beta = 0$ entails perfect alignment of $\theta^\star$ with one of $\mathbf{u}_j, j \in [r]$. As we will see in the result in Theorem 1, the gains provided by multi-step SD are large when $\beta$ is small. In general, both these conditions are somewhat necessary for the separation, as we show in Theorems 2 and 3. We now state our main result. We show that under the above assumption, there exists a family of problem instances, $(\mathbf{X}, \theta^\star, \gamma^2)$, such that the excess risk achieved by $r$-step SD is a factor of $r := \mathrm{rank}(\mathbf{X})$ smaller than that of the ridge estimator *and* the 1-step SD.

**Theorem 1.** *Under the fixed design linear regression in Assumption 1, there exists a family of problem instances satisfying Assumption 2 such that for any instance $(\mathbf{X}, \theta^\star, \gamma^2)$ in the family, it holds that*

$$\exists \lambda > 0, \exists \xi^{(r)} \in \mathbb{R}^r, \quad \mathrm{ExcessRisk}\left(\hat{\theta}(\lambda, \xi^{(r)})\right) \leq \frac{\gamma^2}{n}\left(1 + \beta \frac{\|\theta^\star\|^2 s_1^2}{\gamma^2}\right) , \tag{9}$$

$$\forall \lambda > 0, \forall \xi \in \mathbb{R}, \quad \mathrm{ExcessRisk}\left(\hat{\theta}(\lambda, \xi)\right) \geq \left(\frac{0.99}{2^{11}}\right)(1-\beta)\frac{r\gamma^2}{n} , \text{ and} \tag{10}$$

$$\forall \lambda > 0, \quad \mathrm{ExcessRisk}\left(\hat{\theta}(\lambda)\right) \geq 0.98\left(\frac{1-\beta}{1-0.99\beta}\right)^2\frac{r\gamma^2}{n} , \tag{11}$$

*where $r := \mathrm{rank}(\mathbf{X})$, $n$ is the number of samples, $\hat{\theta}(\lambda, \xi^{(r)})$ and $\hat{\theta}(\lambda, \xi)$ are the $r$-step and 1-step SD estimators defined in Eqs. (8) and (4) respectively, and $\hat{\theta}(\lambda)$ is the ridge estimator defined in Eq. (3).*

This theorem captures a general result for the gains of multi-step SD. In particular, the special case of $\beta = 0$ (i.e. $\theta^\star$ being completely aligned with one of the eigenvectors $\mathbf{u}_j, j \in [r]$) presents an $\Omega(r)$ multiplicative separation between the excess risk of $r$-step SD and $\{1, 0\}$-step SD. We provide precise conditions and a proof in Appendix E. Since each $k$-step SD includes $(k-1)$-step SD as a special case with the proper choice of $\xi^{(k)}$, the hyperparameter-tuned excess risk of repeated SD is monotonically non-increasing. However, it is perhaps unexpected that the multiplicative separation between $r$-step SD and 1-step SD can be as large as $\Omega(r)$, demonstrating the gains of repeated SD. Figure 2 illustrates this $\Omega(r)$ multiplicative separation on a synthetic family of problems. Note that $\Omega(d)$ separation can be achieved by choosing the problem instance to have rank $r = d$, at the cost of requiring many more steps of SD. This $\Omega(d)$ factor is the largest multiplicative separation possible with self-distillation, as shown by the fundamental lower bound in Theorem 4 for any pre-conditioning based approach. In general, if $\beta = O\left(\frac{\gamma^2}{\|\theta^\star\|^2 s_1^2}\right)$ (akin to the inverse signal-to-noise ratio), then there exists an $\Omega(r)$ multiplicative separation between $r$-step SD and $\{1, 0\}$-step SD. In our particular construction of the problem instance for Theorem 1, the additional technical condition (i.e. Condition #2 in the detailed theorem statement in Appendix E) effectively translates this to $\beta = O(1/r)$.

**Remark 4.1.** *SD significantly outperforms ridge by primarily reducing the variance. For the lower bound on ridge's excess risk, i.e., Eq. (11), we ignored the bias term and only used the variance term. The repeated SD (Eq. (9)) primarily reduces the variance to improve the excess risk over Eq. (11).*

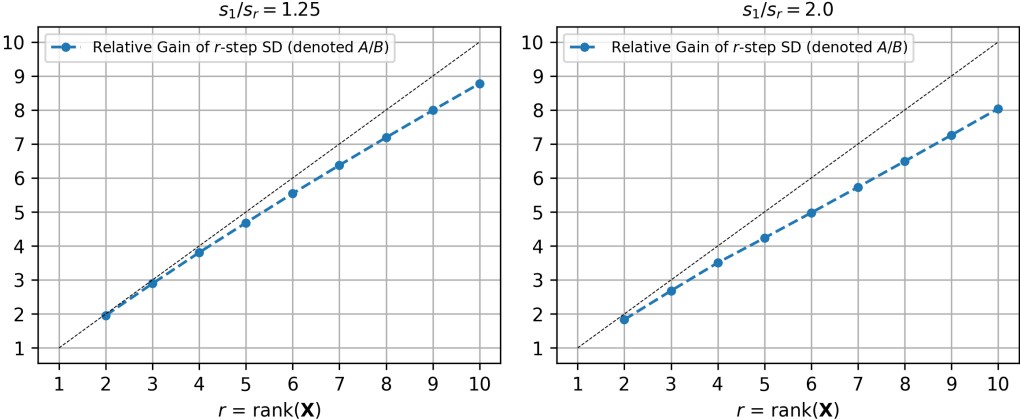

Figure 2: On a synthetic problem family with dimension $d = 100$, noise variance $\gamma = 0.1$, and $\theta^\star = \mathbf{u}_1$ (agreement with Asmp. 2.2); we set the singular values of $\mathbf{X}$ with a power law from $s_1 = 1$ to $s_r = \{0.8, 0.5\}$ (left and right panels) and vary $r = \mathrm{rank}(\mathbf{X})$. Both plots show a linear increase of the relative gain of $r$-step self-distillation in excess risk, i.e. the ratio $A/B$ where $A := \min_{\lambda>0} \mathrm{ExcessRisk}(\hat{\theta}(\lambda))$ and $B := \min_{\lambda>0, \xi^{(r)}\in\mathbb{R}^r} \mathrm{ExcessRisk}(\hat{\theta}(\lambda, \xi^{(r)}))$; demonstrating that $r$-step SD outperforms ridge by a factor of $\Omega(r)$, with the constant inside the $\Omega$ (i.e. slope of the line) changing with the effective condition number, $s_1/s_r$.

## 4.2 Necessity of Assumption 2

In Figure 3, we empirically show on synthetic tasks how violating Assumption 2.1 or 2.2 leads to higher excess risks, even for the $r$-step SD ($r = 4$ in the example). This supports the necessity of both assumptions, which we analytically investigate in the following.

**Necessity of Assumption 2.1 on $\mathbf{X}$.** We assume that the non-zero singular values of $\mathbf{X}$ are unique. This allows us to tightly upper bound the excess risk achieved by $r$-step SD in Eq. (9) via Theorem 4. We show in the following that some version of Assumption 2.1 is also *necessary*. For a more detailed explanation of why we need Assumption 2.1, we refer the reader to Remark 4.2.

**Theorem 2.** *Under the hypotheses of Theorem 1 except for Assumption 2.1, if the singular values of* $\mathbf{X}$ *satisfy* $s_1 = \ldots = s_r = 1$, *where* $r = \mathrm{rank}(\mathbf{X})$, *for all* $k \geq 1$, $\lambda > 0$, *and* $\xi^{(k)} \in \mathbb{R}^k$, *we have*

$$\mathrm{ExcessRisk}\left(\hat{\theta}\left(\lambda, \xi^{(k)}\right)\right) \geq \frac{r\gamma^2}{n}\left(1 + \frac{r\gamma^2}{\sum_{j=1}^r \langle\theta^\star, \mathbf{u}_j\rangle^2}\right)^{-1}. \tag{12}$$

*Furthermore, there exists* $\lambda > 0$ *such that the ridge,* $\hat{\theta}(\lambda)$, *achieves this lower bound with equality.*

We provide a proof in Appendix F. This implies that when there is no gap in the singular values of the input $\mathbf{X}$, there is no separation between ridge and SD estimators (repeated or not). Intuitively, if the $s_j$ are all equal, the pre-conditioner for ridge (i.e., $\mathbf{\Omega}_\lambda^{-1}$) and the pre-conditioner for the repeated SD, both are restricted to have all eigenvalues to be equal. (Repeated) SD has no degrees of freedom to deviate from this. However, $s_j$'s being unequal provides the freedom for the $\xi^{(k)}$ to control the SD's pre-conditioner matrix, and reduce the excess risk. This is also why in Remark 4.2, we hypothesize that numerically, the optimal $(\xi^{(k)})^\star$ depends inversely on the min-gap of the singular values. Figure 4 demonstrates this increasing relationship of the magnitude of the optimal $\xi$ parameters w.r.to the decreasing singular gap.

**Necessity of Assumption 2.2 on $\theta^\star$.** Theorem 1 shows that there is a large separation in the performance of repeated SD over ridge when Assumption 2.2 holds with a small $\beta$. In general, this translates to $\theta^\star$ being highly aligned with *any one* of the eigenvectors of $\mathbf{X}\mathbf{X}^\top$ (not necessarily the leading eigenvector). We show next that if $\theta^\star$ is equally (mis)aligned with all the eigenvectors $\{\mathbf{u}_j\}_{j=1}^r$ of $\mathbf{X}\mathbf{X}^\top$, then again there is no separation between ridge and repeated SD.

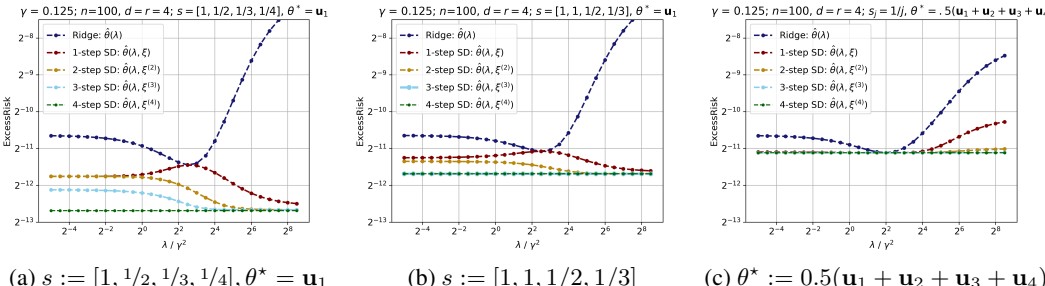

(a) $s := [1, 1/2, 1/3, 1/4], \theta^\star = \mathbf{u}_1$   (b) $s := [1, 1, 1/2, 1/3]$   (c) $\theta^\star := 0.5(\mathbf{u}_1 + \mathbf{u}_2 + \mathbf{u}_3 + \mathbf{u}_4)$

Figure 3: On a synthetic task (explained in Section 5.1), $\mathbf{X}$ has rank 4 with (a) $\theta^\star = \mathbf{u}_1$ and distinct $s_j$'s; (b) $s = [1, 1, 1/2, 1/3]$; (c) $\theta^\star = 0.5(\mathbf{u}_1 + \mathbf{u}_2 + \mathbf{u}_3 + \mathbf{u}_4)$. Each additional step of SD with optimal choice of $\xi^{(k)}$ reduces $\mathrm{ExcessRisk}(\hat{\theta}(\lambda, (\xi^{(k)})^\star))$ for any choice of $\lambda$ on the $x$-axis. Panel (a) satisfies Asmp. 2 and hence 4-step SD is necessary to achieve the optimal excess risk. This is no longer true when Asmp. 2.1 is violated (b) or Asmp. 2.2 is violated (c). Excess risk achieved by 4-step SD (i.e. the green lines) in panels (a) and (c) exactly match the numerical value given by RHS of eq. (14), i.e. the fundamental lower bound for any SD estimator. But this is not the case in panel (b) [which has the same lower bound from eq. (14) as panel (a)], because Asmp. 2.1 is violated.

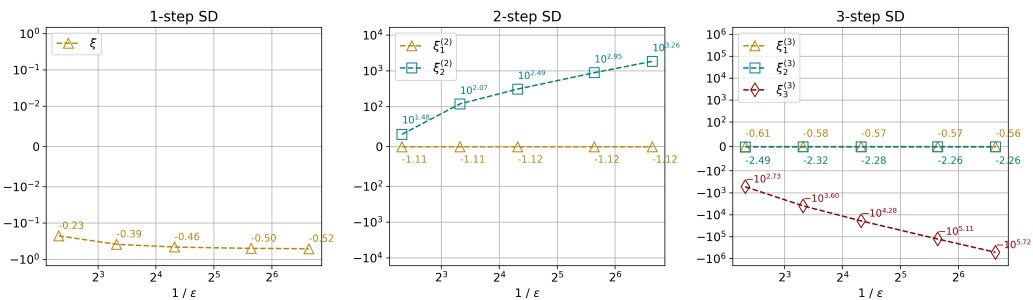

Figure 4: On the synthetic problem from Figure 3a, we fix $\lambda = 0.125$ and set the singular values of $\mathbf{X}$ as $s_j = \{1 - (j-1)\epsilon\}$, i.e. consecutive values are separated by $\epsilon$. For $k$-step SD with $k = \{1, 2, 3\}$, we plot $(\xi^{(k)})^\star(\lambda)$ (i.e. optimal values of the $\xi$ parameters) by varying $\epsilon \in \{0.2, 0.1, 0.05, 0.02, 0.01\}$. The magnitude of $\xi_k^{(k)}$ values increases as the singular gap $\epsilon$ decreases, verifying Remark 4.2.

**Theorem 3.** *Under the hypotheses of Theorem 1 except for Assumption 2.2, if the true parameter $\theta^\star$ satisfies $\langle \theta^\star, \mathbf{u}_j \rangle^2 = z$ for all $j \in [r]$, it holds that for all $z > 0$, $k \geq 1$, $\lambda > 0$, and $\xi^{(k)} \in \mathbb{R}^k$,*

$$\mathrm{ExcessRisk}\left(\hat{\theta}(\lambda, \xi^{(k)})\right) \geq \frac{\gamma^2}{n} \sum_{j=1}^{r} \left(1 + \frac{\gamma^2}{z s_j^2}\right)^{-1}. \tag{13}$$

*Furthermore, there exists $\lambda > 0$ such that the ridge, $\hat{\theta}(\lambda)$, achieves this lower bound with equality.*

We provide a proof in Appendix G. Similar conditions are needed when analyzing 1-step SD in [9] as well; [9, Eq. (9) in Theorem 3.8] is required for the 1-step SD to strictly outperform ridge. Observe that Eq. (9) is violated when either $(i)$ $s_j$'s are all equal or $(ii)$ $\langle \theta^\star, \mathbf{u}_j \rangle^2$'s are all equal.

### 4.3 $r$ steps of self-distillation are sufficient

For a given problem instance $(\mathbf{X}, \theta^\star, \gamma^2)$, the excess risk achieved by the $k$-step SD with parameter $\xi^{(k)}$ can be exactly characterized (Theorem 5), but it is complicated and can only be evaluated numerically in general. On the other hand, we show that there exists a fundamental lower bound that holds for a linear family of estimators including all repeated SD, and this lower bound has a simple characterization (Lemma 4.1). Furthermore, we show that under a mild assumption on the eigenvalues of $\mathbf{X}\mathbf{X}^\top$ in Assumption 2.1, the $r$-step SD achieves the lower bound (Theorem 4). This allows a precise characterization of the performance of $r$-step SD.

**Theorem 4.** *Under the fixed design linear regression in Assumption 1, the excess risk of any $k$-step SD estimator on an instance $(\mathbf{X}, \theta^\star, \gamma^2)$, is lower bounded for all $k \geq 1$, $\lambda > 0$, and $\xi^{(k)} \in \mathbb{R}^k$ by*

$$\text{ExcessRisk}\left(\hat{\theta}(\lambda, \xi^{(k)})\right) \geq \frac{\gamma^2}{n} \sum_{j=1}^r \frac{\langle \theta^\star, \mathbf{u}_j \rangle^2}{\left(\langle \theta^\star, \mathbf{u}_j \rangle^2 + \frac{\gamma^2}{s_j^2}\right)} , \tag{14}$$

*where $(s_j, \mathbf{u}_j)$ is the $j^{th}$ eigenvalue and eigenvector of $\mathbf{X}$ and $r := \text{rank}(\mathbf{X})$. Furthermore, if Assumption 2.1 holds then there exists $\lambda > 0$ and $\xi^{(r)} \in \mathbb{R}^r$ such that the equality is achieved by the $r$-step SD estimator $\hat{\theta}(\lambda, \xi^{(r)})$.*

**Proof sketch.** *(Proof in Appendix H).* The lower bound in Eq. (14) is an instantiation of Lemma 4.1, since $\hat{\theta}(\lambda, \xi^{(k)})$ is a specific linear family estimator with $\mathbf{P} = \mathbf{P}\left(\lambda, \xi^{(k)}\right) \mathbf{\Omega}_\lambda^{-1}$ defined in Eq. (8). To show achievability, we need to show that $\mathbf{P}\left(\lambda, \xi^{(k)}\right) \mathbf{\Omega}_\lambda^{-1} = \mathbf{P}^\star$ for some value of $k$, $\lambda$, and $\xi^{(k)}$. This holds when the below system of $r$ linear equations admits a solution for the $k$ parameters (i.e. $\xi^{(k)}$), with an *extra* free parameter $\lambda > 0$. We show that with $k = r$ and under Assumption 2.1, there exists $\lambda > 0$ that will ensure the existence of a solution for this system of equations.

$$\left(1 - \sum_{i=1}^k \bar{\xi}_i^{(k)} \left\{ 1 - \left(\frac{s_j^2}{\lambda + s_j^2}\right)^i \right\}\right) \frac{s_j^2}{\lambda + s_j^2} = \frac{\langle \theta^\star, \mathbf{u}_j \rangle^2}{\langle \theta^\star, \mathbf{u}_j \rangle^2 + \frac{\gamma^2}{s_j^2}} \qquad \forall j \in [r] \tag{15}$$

**Remark 4.2** (Necessity of Assumption 2.1). *This assumption is required for (15). Otherwise, the LHS would be the same for indices $j$ and $j + 1$ if $s_j = s_{j+1}$, but the RHS could still be different as $\langle \theta^\star, \mathbf{u}_j \rangle \neq \langle \theta^\star, \mathbf{u}_{j+1} \rangle$ generally. If Assumption 2.1 does not hold, there might not be any $\xi^{(k)}$ satisfying the set of equations for a general $\theta^\star \in \mathbb{R}^d$. Further, the system of linear equations in Eq. (15) becomes more ill-conditioned as the singular values $s_j, j \in [r]$ get closer to each other. Capturing this dependence explicitly is outside the scope of this paper.*

**Lower bound for a linear family.** Consider a linear family of estimators of the form $\hat{\theta}(\mathbf{P}) := \mathbf{P} \cdot \mathbf{X} \mathbf{Y}$, for $\mathbf{P} := \mathbf{U}_d \tilde{S} \mathbf{U}_d^\top$, whose eigenspace coincides with that of $\mathbf{X} \mathbf{X}^\top$ (i.e., $\mathbf{U}_d = [\mathbf{u}_1, \ldots, \mathbf{u}_d]$) and has $d$ degrees of freedom represented by the eigenvalues $\tilde{S} = \text{diag}[\tilde{s}_1, \cdots, \tilde{s}_d]$. This is a generic form of any linear estimator, albeit with the restriction of the eigenvectors matching the underlying $\mathbf{U}_d$. In particular, $k$-step SD is an instantiation of this with $\mathbf{P} = \mathbf{P}(\lambda, \xi^{(k)}) \mathbf{\Omega}_\lambda^{-1}$ (refer to Eq (8)).

**Lemma 4.1.** *The Excess Risk for $\hat{\theta}(\mathbf{P}) = \mathbf{P} \cdot \mathbf{X} \mathbf{Y}$ where $\mathbf{P} := \mathbf{U}_d \tilde{S} \mathbf{U}_d^\top$, satisfies*

$$\text{ExcessRisk}\left(\hat{\theta}(\mathbf{P})\right) \geq \frac{\gamma^2}{n} \sum_{j=1}^r \frac{\langle \theta^\star, \mathbf{u}_j \rangle^2}{\left(\langle \theta^\star, \mathbf{u}_j \rangle^2 + \frac{\gamma^2}{s_j^2}\right)} , \tag{16}$$

*with equality achieved at $\mathbf{P} = \mathbf{P}^\star = \mathbf{U}_d \tilde{S}^\star \mathbf{U}_d^\top$, given by*

$$\tilde{s}_j^\star = \begin{cases} \frac{\langle \theta^\star, \mathbf{u}_j \rangle^2}{(\langle \theta^\star, \mathbf{u}_j \rangle^2 s_j^2 + \gamma^2)} , & j \leq r \text{ (i.e., } s_j > 0) \\ \text{any real value} , & j \geq r + 1 \text{ (i.e., } s_j = 0) \end{cases} . \tag{17}$$

**Proof sketch.** *(Proof in Appendix H.1).* One can expand the excess risk for $\hat{\theta}(\mathbf{P})$, which is a quadratic expression in $\tilde{s}_j, j \in [d]$. Completing the squares gives the result immediately.

## 4.4 The excess risk for the $k$-step SD estimator is quadratic in $\bar{\xi}^{(k)} \in \mathbb{R}^k$

We give an explicit formula for the excess risk achieved by for the $k$-step SD estimator from Eq. (8). Since $\hat{\theta}(\lambda, \xi^{(k)})$ is linear in each of $\bar{\xi}_i^{(k)}, i \in [k]$ (recall that $\bar{\xi}^{(k)}$ is a reparametrization of $\xi^{(k)}$), the overall excess risk is *quadratic* in $\bar{\xi}^{(k)}$ as shown below. Appendix I provides a proof and the expressions for $M^{(k)}$, $m^{(k)}$, and $c$. This quadratic form will be especially useful in experiments.

**Theorem 5** (Informal version of Theorem 7 in Appendix I). *Under the fixed design linear regression in Assumption 1, the excess risk achieved by the $k$-step SD is quadratic in $\bar{\xi}^{(k)} \in \mathbb{R}^k$:*

$$\text{ExcessRisk}\left(\hat{\theta}(\lambda, \xi^{(k)})\right) = \left(\bar{\xi}^{(k)}\right)^\top \underbrace{M^{(k)}}_{\in \mathbb{R}^{k \times k}} \left(\bar{\xi}^{(k)}\right) + 2\left(\bar{\xi}^{(k)}\right)^\top \underbrace{m^{(k)}}_{\in \mathbb{R}^k} + c . \tag{18}$$

From the detailed expressions given in Appendix I, we note that $M^{(k)}$ is a sum of $r$ symmetric rank-1 matrices, which means it can have a maximum rank of $r$. This implies that $M^{(k)} \in \mathbb{R}^{k \times k}$ for $k > r$ is rank-deficient (causing no additional decrease in the excess risk if the $\bar{\xi}^{(r)} \in \mathbb{R}^r$ were chosen optimally to minimize the excess risk). This indicates that $r$ steps of SD might be sufficient to achieve the optimal excess risk, which is indeed what we observe in Theorem 4.

## 5 Experiments

In this section, we empirically show that multi-step SD can outperform the ridge and single-step SD. We first present a synthetic setting (section 5.1) to validate our theory. In section 5.2, we discuss a strategy to select $\xi$ parameters based on the theoretical insight from section 4.4. In section 5.3, we implement that strategy on real-world regression tasks and show that it can indeed select performant $\xi$ values that provide multi-step SD estimators that achieve a smaller test risk.

### 5.1 Synthetic Experiments

We validate our theoretical results with a fixed design synthetic experiment. We consider a problem with $d = r = 4$, and set problem parameters $(\mathbf{X}, \theta^\star, \gamma^2)$. Namely, $\mathbf{X}$'s singular values are set as $s_j := 1/j$ for $j \in [4]$, and $\theta^\star := \mathbf{u}_1$ as in Theorem 1. Figure 3 shows the result for $\gamma = 0.125$, along with two more settings that validate the necessity of our assumptions (validating Theorems 2 and 3). Figure 3a confirms that repeated steps of SD do provide a reduction in the excess risk, since the *lowest point* of the curve for each $k$ reduces as $k$ increases. Also note that the optimal $\lambda$ for each $k$ (one that produces lowest excess risk estimator) is different. Appendix J presents some more settings, including $\theta^\star := 1/\sqrt{2}(\mathbf{u}_1 + \mathbf{u}_2)$ for comparison with [9]. An interesting phenomenon in Figure 3 is that local maxima in $k$-step SD's curve coincide with local minima in $(k-1)$-step SD's curve, which was proven for $k = 1$ in [9], and we observe empirically for all values of $k$.

**Explanation of Figures 3, 5**. For the fixed design synthetic experiment in Figure 3 and the random design real-world experiment in Figure 5 (section 5.3), the curves plotted are with the optimal $(\xi^{(k)})^\star$ for each $\lambda$. Hence, the curve of $k$-step SD will point-wise be lower/equal to the curve of $(k-1)$-step SD, since more steps of SD only provide more freedom. We say $k$-step SD *strictly* dominates $(k-1)$-step SD when the minimum value of $k$-step SD's excess risk is strictly lower than that of $(k-1)$-step SD. For Figure 3, the optimal $(\xi^{(k)})^\star$ is found analytically from the problem parameters. For real-world datasets in Figure 5, we use the strategy in section 5.2 to find $(\xi^{(k)})^\star$.

### 5.2 Choosing the hyperparameters $\xi$ for real-world datasets

We have shown that at the cost of introducing additional hyperparameters *and* setting them to their optimal values, one can extract a large (upto $\Omega(d)$) performance gain. However, how does one select these $\xi$'s for real-world datasets? The standard practice is to use a validation set, and perform a grid search. But this becomes infeasible for $k$-step SD for larger values of $k$, since performing a search over parameters in $\mathbb{R}^{k+1}$ (i.e $k$ values of $\xi^{(k)}$ and 1 value of $\lambda$) quickly becomes impractical. However, our theoretical analysis provides an insight that can be used to directly compute the optimal $\xi^{(k)} \in \mathbb{R}^k$ (for a chosen $\lambda$) given a few evaluations on the validation set with certain chosen $\xi^{(k)}$ values. Namely, Theorem 5 tells us that the ExcessRisk is quadratic in $\bar{\xi}^{(k)}$ (the reparameterized version). Now the coefficients of the quadratic depend on unknown quantities (like $\theta^\star, \gamma^2$), however just knowing the quadratic nature of the functional, we can use the validation set to estimate these coefficients. To estimate the coefficients for $k$-step SD, we need $k(k+3)/2 + 1$ evaluations on the validation set. Appendix K provides more discussion, and a detailed illustration of the above process for $k = 1, 2$. Note that this is feasible when the cost/time needed for a single training run of a $k$-step SD is small (since we need to perform it $O(k^2)$ times), which holds true for linear regression.

### 5.3 Real-world regression experiments

We implement multi-step SD for real-world regression tasks from the UCI repository [18], and demonstrate that 2-step SD can outperform ridge and 1-step SD. Note that for this section, the test set will contain fresh samples of $X \in \mathbb{R}^d$, i.e. random design linear regression instead of fixed design. Our metric for an estimator's performance will now be mean squared error (MSE) on a test set of unseen examples, which is the empirical version of total risk (i.e. excess risk translated by the unknown noise variance $\gamma^2$). Using the training and validation splits, we compute $(i)$ Optimal ridge: $\hat{\theta}(\lambda_0^\star)$, $(ii)$ Optimal 1-step SD: $\hat{\theta}(\lambda_1^\star, \xi^\star)$, and $(iii)$ Optimal 2-step SD: $\hat{\theta}(\lambda_2^\star, (\xi_1^\star, \xi_2^\star))$. The procedure is to plot the MSE on the validation set for a grid of $\lambda$ values for all three estimators, and

choose the $\lambda$ that achieves the lowest error for each one (for any given $\lambda$, the optimal $\xi^\star(\lambda)$ is chosen by the strategy described in Section 5.2). Finally, we evaluate the MSE of all three selected estimators (i.e. with the chosen optimal hyperparameters) on the test set, which serves as our performance metric (refer to Table 1). Appendix L explains the overall methodology in greater detail. We apply this methodology on three datasets (dataset descriptions in Appendix L.1).

Table 1 describes the results we observe. For two of the three datasets, 2-step SD can outperform both ridge and 1-step SD. For the Air Quality dataset, 2-step SD significantly outperforms both ridge and 1-step SD, reducing the MSE by $47.2\%$ compared to the optimal ridge. In contrast, for the AEP dataset, we observe that the SD process cannot improve upon the ridge at all. The MSE curves in Figure 5 also shed light on these observations. Notice how Figures 5a, 5b show a gap in the ridge and 2-step SD (similar to Figure 3a), whereas Figure 5c shows no such gap (similar to Figure 3c).

In Appendix L.2, we explain the lack of gains from self-distillation on the AEP dataset from the lens of Assumption 2.2. Recall that Theorem 1 says that the gains of repeated SD are most prominent with a small $\beta$ in Assumption 2.2. Figure 10 shows that the $\theta^\star$ for the AEP dataset is at most $\sim 35\%$ aligned with any of the eigenbasis directions, corresponding to a $\beta \approx 0.65$. The same alignment for the other two datasets is $\sim 80\%$, corresponding to a much smaller $\beta \approx 0.2$. In Appendix L.3, we also verify that the strategy described in section 5.2 indeed selects performant $\xi$ values.

Table 1: Chosen hyperparameter values and the achieved test set MSE for ridge and $1, 2$-step SD.

| Dataset | | Optimal ridge | Optimal 1-step SD | Optimal 2-step SD |
|---|---|---|---|---|
| `Air Quality` | Optimality hyperparameters | $\lambda_0^\star = 10^2$ | $\lambda_1^\star, \xi^\star = 10^3, -4.1$ | $\lambda_2^\star, (\xi_1^\star, \xi_2^\star) = 10^3, (-0.9, -16.2)$ |
| | Test set MSE | 2.01 | 1.99 | **1.06** |
| `Airfoil` | Optimality hyperparameters | $\lambda_0^\star = 10^2$ | $\lambda_1^\star, \xi^\star = 10^0, 66.5$ | $\lambda_2^\star, (\xi_1^\star, \xi_2^\star) = 10^3, (-1.8, -7.8)$ |
| | Test set MSE | 1.34 | 1.22 | **1.19** |
| `AEP` | Optimality hyperparameters | $\lambda_0^\star = 10^{2.5}$ | $\lambda_1^\star, \xi^\star = 10^{2.5}, 0.1$ | $\lambda_2^\star, (\xi_1^\star, \xi_2^\star) = 10^{2.5}, (-2.4, -2.3)$ |
| | Test set MSE | **0.62** | 0.62 | 0.63 |

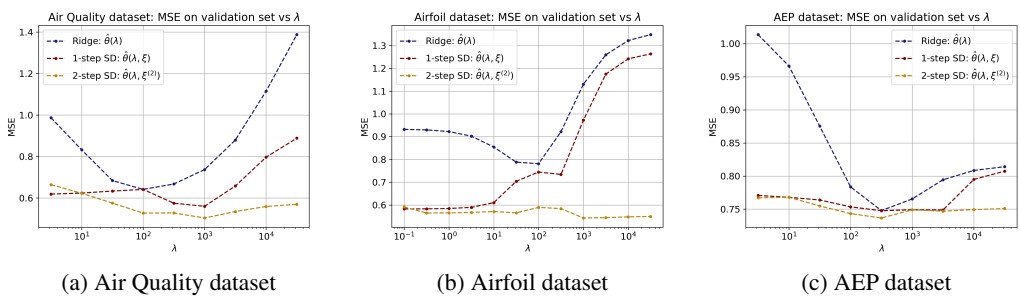

| (a) Air Quality dataset | (b) Airfoil dataset | (c) AEP dataset |
|---|---|---|

Figure 5: Validation set MSE vs $\lambda$ for three estimators: Ridge, 1-step SD and 2-step SD.

## 6  Conclusion and Broader Impacts

In this paper, we theoretically studied the multi-step self-distillation for fixed design linear regression, with the goal of characterizing its performance compared to the single-step SD. Perhaps surprisingly, we demonstrated that the optimal multi-step SD can outperform the optimal single-step SD by a factor as large as $d$ in the estimator's excess risk, where $d$ is the input dimension of the regression. Our analysis is limited by the fixed design assumption, and it would be useful to study the case of random design linear regression as well. We empirically demonstrated the gains from using 2-step SD on simple linear regression tasks. Larger scale empirical studies of multi-step SD, especially leveraging the insights of Section 5.2 on hyperparameter search, remain as a direction of future work.

Our contributions are largely on the theoretical understanding of multi-step self-distillation, and its potential performance gains. At a high-level, self-distillation can use data more effectively, since it allows us to extract more knowledge from the same training dataset. In today's age with data being one of the most important resources, this has positive potential impacts through more judicious use of data. On the other hand, we propose to use multiple steps of self-distillation, requiring more compute and potentially contributing to higher environmental costs.

## Acknowledgements

We thank Raghav Somani for numerous insightful discussions at various stages of this project. This work is supported by NSF grants no. 2019844, 2112471, 2229876, 2134106, 2143493, 2134012, and 2229881; and Microsoft Grant for Customer Experience Innovation.

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

## A    Notation

In this short section, we collect some notation used throughout the proofs.

**Decomposition of X**. Let $\text{rank}(\mathbf{X}) = r$ and the SVD of $\mathbf{X}$ be $\mathbf{X} = \sum_{j=1}^{r} s_j \mathbf{u}_j \mathbf{v}_j^T$ where $s_1 \geq s_2 \geq \cdots \geq s_r > 0$, and each $\mathbf{u}_j \in \mathbb{R}^d$ and $\mathbf{v}_j \in \mathbb{R}^n$. Further, let $\{\mathbf{u}_1, \mathbf{u}_2, \cdots, \mathbf{u}_d\}$ be the full set of left singular vectors of $\mathbf{X}$ (even those corresponding to zero singular values), forming an orthonormal basis of $\mathbb{R}^d$. Let $\mathbf{U}_d \in \mathbb{R}^{d \times d}$ and $\mathbf{U}_r \in \mathbb{R}^{d \times r}$ denote the left singular matrix of $\mathbf{X}$ for the full and truncated set of left singular vectors respectively. Similarly, let $\mathbf{V}_d \in \mathbb{R}^{n \times d}$ and $\mathbf{V}_r \in \mathbb{R}^{n \times r}$ denote the full and truncated right singular matrix. Let $S_d \in \mathbb{R}^{d \times d}$ and $S_r \in \mathbb{R}^{r \times r}$ denote the collection of the singular values (with and without zeros respectively). Then, it holds that

$$\mathbf{X} = \mathbf{U}_r S_r \mathbf{V}_r^\top = \mathbf{U}_d S_d \mathbf{V}_d^\top \ . \tag{19}$$

**Indices**. Throughout the text, the indices $i$ lie in $[k]$, i.e. they denote subsequent steps of self-distillation. The indices $j$ lie in $[r]$ or $[d]$, ie they denote dimensions of the $d$-dimensional space (aligned with vectors of $\mathbf{U}_r$ or $\mathbf{U}_d$). $v_j$, $V_{i,j}$ will denote indexing into a vector $v$, matrix $V$. There is one exception to $v_j$ denoting a vector's $j^{th}$ element, which is the below.

**Components of $\theta^\star$ on $\mathbf{U}_d$**. We will denote $\theta_j^\star := \langle \theta^\star, \mathbf{u}_j \rangle^2$, $j \in [d]$ as the components of $\theta^\star$ onto $\mathbf{X}$'s left singular space. Note that $\sum_{j=1}^{d} \theta_j^\star = \|\theta^\star\|_2^2$.

## B    Discussion on norm used in excess risk metric

We point out that Das and Sanghavi [9] used the $\|.\|_2$ norm instead of the more natural $\|.\|_{\hat{\Sigma}_n}$ norm to measure their fixed design excess risk (in eq (2)). Almost all our results have an equivalent version in the $\|.\|_2$ norm setting also, since the only difference is in the relative weighing of the underlying dimensions of variation, i.e. with the $\|.\|_{\hat{\Sigma}_n}$ norm, $\forall j \in [d]$, direction $\mathbf{u}_j$ is weighed by $s_j^2/n$ instead of a constant 1 weight (independent of $j$). In particular, we also present the version of [9, Eq. (9) from Theorem 3.8] that will result in a strict dominance like Eq. (7) under the $\hat{\Sigma}_n$ norm, i.e.

$$\min_{\lambda \geq 0, \xi \in \mathbb{R}} \underbrace{\mathbb{E}_{\boldsymbol{\eta}} \left[ \|\hat{\theta}(\lambda, \xi) - \theta^\star\|_{\hat{\Sigma}_n}^2 \right]}_{=\text{ExcessRisk}(\hat{\theta}(\lambda, \xi))} < \min_{\lambda \geq 0} \underbrace{\mathbb{E}_{\boldsymbol{\eta}} \left[ \|\hat{\theta}(\lambda) - \theta^\star\|_{\hat{\Sigma}_n}^2 \right]}_{=\text{ExcessRisk}(\hat{\theta}(\lambda))} \ . \tag{20}$$

**Proposition B.1.** *Let $\lambda^\star := argmin_{\lambda > 0} \text{ExcessRisk}(\hat{\theta}(\lambda))$. Then Eq. (20) holds on a problem instance $(\mathbf{X}, \theta^\star, \gamma^2)$ when*

$$\sum_{k=1}^{r} \sum_{j=1}^{k-1} \frac{s_j^4 s_k^4 \left( s_j^2 - s_k^2 \right) \left( \langle \theta^\star, \mathbf{u}_k \rangle^2 - \langle \theta^\star, \mathbf{u}_j \rangle^2 \right)}{\left( \lambda^\star + s_j^2 \right)^4 \left( \lambda^\star + s_k^2 \right)^4} < 0 \ . \tag{21}$$

Note that this differs from [9, Eq. (9)] in just one respect: it has $s_j^4 s_k^4$ instead of $s_j^2 s_k^2$.

## C    Details on $\xi$ parameters for general $k$-step SD

### C.1    Full $k$-step SD is representationally no larger than Repeated $k$-step SD

We first illustrate the Full $k$-step SD in Figure 6. The repeated version introduces $k$ extra hyperparameters in the form of $\xi^{(k)} \in \mathbb{R}^k$ parameters, whereas the full version introduces $k(k+1)/2$.

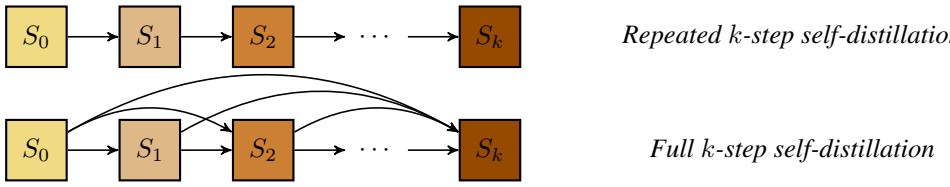

Figure 6: Illustrating two possible generalizations of 1-step SD to a $k$-step process.

Consider the case of $k = 2$, since that is the lowest value of $k$ for which the full version and the repeated version differ. Figure 7 illustrates this difference explicitly. We will show that the freedom

of $\tilde{\xi} \in \mathbb{R}^3$ is no more than the freedom of $\xi \in \mathbb{R}^2$. This shows the equivalence of the full 2-step and the repeated 2-step versions, when $\tilde{\xi} \in \mathbb{R}^3$, $\xi \in \mathbb{R}^2$ are free parameters optimized over the entire respective spaces. Such equivalence for the general $k$-step case is then easy to see.

Nodes $S_0, \tilde{S}_0$ are both solving the ridge regression problem (3). Let $\theta_0 = \tilde{\theta}_0 = \mathbf{\Omega}_\lambda^{-1}\mathbf{X}\mathbf{Y}$ denote the estimator for both these nodes. Similarly $S_1$ and $\tilde{S}_1$ are solving the same problem, although with different parameters. We have $\theta_1(\xi_1) = \mathbf{\Omega}_\lambda^{-1}\mathbf{X}\left(\xi_1 \cdot \mathbf{X}^\top\bar{\theta}_0 + (1 - \xi_1) \cdot \mathbf{Y}\right)$ (and similarly, one can write $\tilde{\theta}_1$ with $\tilde{\xi}_1$ instead of $\xi_1$). Now the node $S_2$ is also solving a 1 parameter supervised SD problem, so $\theta_2(\xi_1, \xi_2) = \mathbf{\Omega}_\lambda^{-1}\mathbf{X}\left(\xi_2 \cdot \mathbf{X}^\top\theta_1(\xi_1) + (1 - \xi_2) \cdot \mathbf{Y}\right)$. Expanding this gives

$$\theta_2(\xi_1, \xi_2) = \left\{(1 - \xi_2) \cdot \mathbf{I}_d + (\xi_2 - \xi_1\xi_2) \cdot \mathbf{\Omega}_\lambda^{-1}\mathbf{X}\mathbf{X}^\top + \xi_1\xi_2 \cdot (\mathbf{\Omega}_\lambda^{-1}\mathbf{X}\mathbf{X}^\top)^2\right\}\mathbf{\Omega}_\lambda^{-1}\mathbf{X}\mathbf{Y} \ . \quad (22)$$

But the optimization problem for $\tilde{S}_2$ is a 2 parameter supervised SD. It evaluates to

$$\text{argmin}_{\theta \in \mathbb{R}^d}\left(\frac{\tilde{\xi}_{2a}}{2}\|\mathbf{X}^\top\tilde{\theta}_0 - \mathbf{X}^\top\theta\|^2 + \frac{\tilde{\xi}_{2b}}{2}\|\mathbf{X}^\top\tilde{\theta}_1 - \mathbf{X}^\top\theta\|^2 + \frac{(1 - \tilde{\xi}_{2a} - \tilde{\xi}_{2b})}{2}\|\mathbf{Y} - \mathbf{X}^\top\theta\|^2 + \frac{\lambda}{2}\|\theta\|^2\right) \ .$$

Following through a similar calculation, we observe that $\tilde{\theta}_2$ for node $\tilde{S}_2$ is given by

$$\tilde{\theta}_2(\tilde{\xi}_1, \tilde{\xi}_{2a}, \tilde{\xi}_{2b}) = \left\{(1 - \tilde{\xi}_{2a} - \tilde{\xi}_{2b}) \cdot \mathbf{I}_d + (\tilde{\xi}_{2a} + \tilde{\xi}_{2b} - \tilde{\xi}_1\tilde{\xi}_{2b}) \cdot \mathbf{\Omega}_\lambda^{-1}\mathbf{X}\mathbf{X}^\top + \tilde{\xi}_1\tilde{\xi}_{2b} \cdot (\mathbf{\Omega}_\lambda^{-1}\mathbf{X}\mathbf{X}^\top)^2\right\}\mathbf{\Omega}_\lambda^{-1}\mathbf{X}\mathbf{Y} \ .$$
$$(23)$$

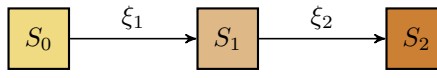

Repeated 2-step self-distillation

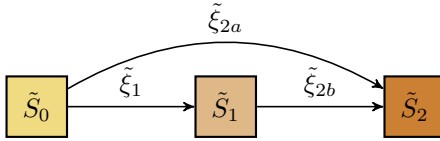

Full 2-step self-distillation

Figure 7: Repeated vs Full 2-step SD. We show that the extra freedom of the parameter $\tilde{\xi}_{2a}$ does not provide any additional freedom, when the other two $\tilde{\xi}_1, \tilde{\xi}_{2b}$ are optimized as free parameters.

Equations (22) vs (23) show that the full 2-step offers the same freedom as the repeated 2-step. However the repeated version has one shortcoming. To generate the optimal 2-step SD estimator, $\xi_1$ needs to be different than the value needed for generating the optimal 1-step SD estimator. That is, the repeated $k$-step version, with its $k$ free parameters, allows only to generate the *final $k^{th}$*-step estimator as the optimal one (i.e. if we choose the $(\xi_1, \xi_2)$ values so that the $2^{nd}$ estimator is the optimal 2-step SD estimator, then the $1^{st}$ estimator with the chosen $\xi_1$ won't be the optimal 1-step SD estimator). Whereas the full $k$-step version, with all its $k(k + 1)/2$ free parameters, allows us to generate a sequence of *all $k$* optimal estimators.

## C.2 Proof that $k$-step SD estimator with $\xi^{(k)} \in \mathbb{R}^k$ will have the form given in eq (8)

Consider Figure 1. $\xi^{(k)} \in \mathbb{R}^k$ is the set of actual imitation parameters used for running $k$-step (repeated) SD. Let $\hat{\theta}(\lambda, \xi^{(k)})$ denote the $k$-step estimator generated by using $\xi^{(k)}$ parameters. In what follows, we will prove that $\hat{\theta}(\lambda, \xi^{(k)})$ will have the form given in eq (8), with $\bar{\xi}^{(k)} \in \mathbb{R}^k$ as the described reparametrization of $\xi^{(k)} \in \mathbb{R}^k$. Since we're in the repeated version, the $k^{th}$ step objective will be a combination of losses w.r.to ground-truth labels and predictions from the $(k - 1)^{th}$ step. So, the objective for the $k^{th}$ step is

$$\hat{\theta}(\lambda, \xi^{(k)}) := \text{argmin}_{\theta \in \mathbb{R}^d}\left(\frac{\xi_k^{(k)}}{2}\|\mathbf{X}^\top\hat{\theta}(\lambda, \xi^{(k-1)}) - \mathbf{X}^\top\theta\|^2 + \frac{(1 - \xi_k^{(k)})}{2}\|\mathbf{Y} - \mathbf{X}^\top\theta\|^2 + \frac{\lambda}{2}\|\theta\|^2\right) \ .$$
$$(24)$$

Note that $\hat{\theta}(\lambda, \xi^{(k)})$ recursively depends on predictions from the previous $\hat{\theta}(\lambda, \xi^{(k-1)})$. This objective is of the form in eq (4), so similar to eq (5), we have the following expression

$$\hat{\theta}(\lambda, \xi^{(k)}) = \left(\mathbf{X}\mathbf{X}^\top + \lambda\mathbf{I}_d\right)^{-1}\mathbf{X}\left\{\xi_k^{(k)} \cdot \mathbf{X}^\top\hat{\theta}(\lambda, \xi^{(k-1)}) + (1 - \xi_k^{(k)}) \cdot \mathbf{Y}\right\} \quad (25)$$

$$= \left\{ (1 - \xi_k^{(k)}) \cdot \mathbf{\Omega}_\lambda^{-1} \mathbf{X} \mathbf{Y} + \xi_k^{(k)} \cdot \mathbf{\Omega}_\lambda^{-1} \mathbf{X} \mathbf{X}^\top \hat{\theta}(\lambda, \xi^{(k-1)}) \right\} \ . \tag{26}$$

Using this, we can inductively prove that the general form of this estimator is captured by Eq. (8).

**Claim**. With the described reparametrization, i.e., $\bar{\xi}_i^{(k)} := (1 - \xi_{k-i}^{(k)}) \prod_{l=k-i+1}^{k} \xi_l^{(k)}$ for each $i \in [k]$ (where we let $\xi_0^{(k)} = 0$), Eq. (8) is the solution to the recursive form Eq. (26).

**Proof**. We will prove this by induction.

*Base Case*. From eqs (8) and (26), the case for $k = 1$ is true (with $\bar{\xi}^{(1)} = \xi^{(1)}$).
*Inductive Step*. Assuming Eq. (8) captures the solution of Eq. (26) for $k - 1$, we get

$$\hat{\theta}(\lambda, \xi^{(k)}) = \left\{ \left\{ 1 - \xi_k^{(k)} \right\} \mathbf{I}_d + \xi_k^{(k)} \mathbf{\Omega}_\lambda^{-1} \mathbf{X} \mathbf{X}^\top \left( \left\{ 1 - \sum_{i=1}^{k-1} \bar{\xi}_i^{(k-1)} \right\} \mathbf{I}_d + \sum_{i=1}^{k-1} \bar{\xi}_i^{(k-1)} \left( \mathbf{\Omega}_\lambda^{-1} \mathbf{X} \mathbf{X}^\top \right)^i \right) \right\} \cdot \mathbf{\Omega}_\lambda^{-1} \mathbf{X} \mathbf{Y} \ .$$

This again satisfies the form in equation (8), when the following coefficients match

$$\xi_k^{(k)} = \sum_{i=1}^{k} \bar{\xi}_i^{(k)} \ ,$$

$$\xi_k^{(k)} \cdot (1 - \sum_{i=1}^{k-1} \bar{\xi}_i^{(k-1)}) = \bar{\xi}_1^{(k)} \ ,$$

$$\xi_k^{(k)} \cdot \bar{\xi}_{i-1}^{(k-1)} = \bar{\xi}_i^{(k)} \qquad \forall i \in \{2, 3, \cdots, k\} \ .$$

One can then see that the described reparametrization makes the above hold true.

**Remark**. Since $\hat{\theta}(\lambda, \xi^{(1)})$ is simply the 1-step SD estimator with the form $\hat{\theta}(\lambda, \xi^{(1)}) = \mathbf{P} \cdot \mathbf{\Omega}_\lambda^{-1} \mathbf{X} \mathbf{Y}$ for some preconditioner $\mathbf{P}$, plugging this in the equation eq (26), we realize that we can factor out $\mathbf{\Omega}_\lambda^{-1} \mathbf{X} \mathbf{Y}$ (i.e. the ridge solution) from the expression on the right side. That is, $\hat{\theta}(\lambda, \xi^{(2)}) = \mathbf{P}' \cdot \mathbf{\Omega}_\lambda^{-1} \mathbf{X} \mathbf{Y}$ for some different pre-conditioner $\mathbf{P}'$. This is why inductively we get that $\hat{\theta}(\lambda, \xi^{(k)})$, $k \geq 1$ all produce a pre-conditioning on the ridge solution, as shown in eq (8). Further, eq (26) also dictates why we get increasing powers of the term $\mathbf{\Omega}_\lambda^{-1} \mathbf{X} \mathbf{X}^\top$ in the final expression.

### C.3 Explicit reparametrization for $k = 2, 3$

We explicitly demonstrate the reparametrization for $k = 2, 3$. As noted in Section 5.2 (and Appendix K), owing to the quadratic form of excess risk in $\bar{\xi}^{(k)}$, one can find the optimal $\bar{\xi}^{(k)}$ analytically. It can then be translated back to the original $\xi^{(k)}$ as follows.

For $k = 2$, the form is (dropping the $.^{(2)}$ for ease)

$$\xi_1 = 1 - \frac{\bar{\xi}_1}{(\bar{\xi}_1 + \bar{\xi}_2)} \quad , \qquad \xi_2 = \bar{\xi}_1 + \bar{\xi}_2 \ . \tag{27}$$

For $k = 3$, the form is (dropping the $.^{(3)}$ for ease)

$$\xi_1 = 1 - \frac{\bar{\xi}_2}{\bar{\xi}_2 + \bar{\xi}_3} \quad , \qquad \xi_2 = 1 - \frac{\bar{\xi}_1}{\bar{\xi}_1 + \bar{\xi}_2 + \bar{\xi}_3} \quad , \qquad \xi_3 = \bar{\xi}_1 + \bar{\xi}_2 + \bar{\xi}_3 \ . \tag{28}$$

## D Algebraic expansion of $\hat{\theta}(\lambda, \xi^{(k)})$ from eq (8)

**Lemma D.1.** *The estimator $\hat{\theta}(\lambda, \xi^{(k)})$ given in eq (8) can be expanded as*

$$\hat{\theta}(\lambda, \xi^{(k)}) = \sum_{j=1}^{d} \left( 1 - \sum_{i=1}^{k} \bar{\xi}_i^{(k)} \left\{ 1 - \left( \frac{s_j^2}{\lambda + s_j^2} \right)^i \right\} \right) \cdot \frac{s_j^2}{\lambda + s_j^2} \cdot \langle \theta^\star, \mathbf{u}_j \rangle \cdot \mathbf{u}_j$$

$$+ \sum_{j=1}^{d} \left( 1 - \sum_{i=1}^{k} \bar{\xi}_i^{(k)} \left\{ 1 - \left( \frac{s_j^2}{\lambda + s_j^2} \right)^i \right\} \right) \cdot \frac{s_j}{\lambda + s_j^2} \cdot \langle \boldsymbol{\eta}, \mathbf{v}_j \rangle \cdot \mathbf{u}_j$$

*Proof.* The expansion relies on $\mathbf{X} \mathbf{X}^\top = \mathbf{U}_d S_d^2 \mathbf{U}_d^\top$ and $\mathbf{X} \mathbf{X}^\top + \lambda \mathbf{I}_d = \mathbf{U}_d (S_d^2 + \lambda \mathbf{I}_d) \mathbf{U}_d^\top$. $\mathbf{U}_d$ is orthonormal ($\mathbf{U}_d \mathbf{U}_d^\top = \mathbf{I}_d = \mathbf{U}_d^\top \mathbf{U}_d$), which neatly cancels all occurences of $\mathbf{U}_d$ in the middle, and allows us to directly combine the matrices in the eigenvalue space. $\qquad \square$

# E  Detailed version and a proof of Theorem 1

We first write a detailed version of the theorem that exactly characterizes the family of problem instances that admits the separation. We then present a proof. We point the reader to Appendix A for notations used throughout the proof.

**Theorem 6** (Detailed version). *Consider the following conditions on $\theta^\star, \mathbf{X}$ that characterize a family of problem instances $\{(\mathbf{X}, \theta^\star, \gamma^2)\}$:*

1. *$\theta^\star$ satisfies $\exists j \in [r], \langle \theta^\star, \mathbf{u}_j \rangle^2 \geq (1-\beta) \cdot \|\theta^\star\|^2$*          *[Assumption 2.2]*

2. *$\|\theta^\star\|^2 \geq 198 \cdot \left(r\gamma^2/s_r^2\right) \cdot \left(s_1/s_r\right)^4$*

3. *Assumption 2.1 holds on the singular values of $\mathbf{X}$*          *[Assumption 2.1]*

4. *$\left(s_1^2/s_r^2\right) \leq 2$ holds on the singular values of $\mathbf{X}$*

*Under Condition #1 + Condition #3, it holds that*

$$\exists \lambda > 0, \exists \xi^{(r)} \in \mathbb{R}^r, \quad \text{ExcessRisk}\left(\hat{\theta}(\lambda, \xi^{(r)})\right) \leq \frac{\gamma^2}{n}\left(1 + \beta \frac{\|\theta^\star\|^2 s_1^2}{\gamma^2}\right) \quad \text{(Rewriting eq (9))}$$

*Under Condition #1 + Condition #2 + Condition #4, it holds that*

$$\forall \lambda > 0, \forall \xi \in \mathbb{R}, \quad \text{ExcessRisk}\left(\hat{\theta}(\lambda, \xi)\right) \geq \frac{0.99}{2^{11}}(1-\beta)\frac{r\gamma^2}{n} \quad \text{(Rewriting eq (10))}$$

*Under Condition #1 + Condition #2, it holds that*

$$\forall \lambda > 0, \quad \text{ExcessRisk}\left(\hat{\theta}(\lambda)\right) \geq 0.98\left(\frac{1-\beta}{1-0.99\beta}\right)^2 \frac{r\gamma^2}{n} \quad \text{(Rewriting eq (11))}$$

*Proof.* We will analyze all three: $k$-step SD, ridge, and 1-step SD.

Let $j' \in [r]$ be the index that corresponds to the maximum (unsigned) alignment between $\theta^\star$ and $\mathbf{u}_j$, among all $j \in [r]$. If it is not unique, pick any of the corresponding indices. That is, $j'$ satisfies $\langle \theta^\star, \mathbf{u}_{j'} \rangle^2 \geq \langle \theta^\star, \mathbf{u}_j \rangle^2$ for all $j \in [r]$. Then, Condition #1 translates to the below two inequalities,

$$\theta_{j'}^\star \equiv \langle \theta^\star, \mathbf{u}_{j'} \rangle^2 \geq (1-\beta) \cdot \|\theta^\star\|^2 \quad , \tag{29}$$

$$\sum_{j=1, j \neq j'}^{r} \theta_j^\star \equiv \sum_{j=1, j \neq j'}^{r} \langle \theta^\star, \mathbf{u}_j \rangle^2 \leq \sum_{j=1, j \neq j'}^{d} \langle \theta^\star, \mathbf{u}_j \rangle^2 \leq \beta \cdot \|\theta^\star\|^2 \quad . \tag{30}$$

$k$**-step SD**: Since Assumption 2.1 holds, from Theorem 4 we directly have

$$\exists \lambda > 0, \exists \xi^{(r)} \in \mathbb{R}^r, \quad \text{ExcessRisk}\left(\hat{\theta}\left(\lambda, \xi^{(r)}\right)\right) = \frac{\gamma^2}{n}\sum_{j=1}^{r}\frac{\theta_j^\star}{\left(\theta_j^\star + \frac{\gamma^2}{s_j^2}\right)} \tag{31}$$

$$= \frac{\gamma^2}{n}\left(\frac{\theta_{j'}^\star}{\left(\theta_{j'}^\star + \frac{\gamma^2}{s_{j'}^2}\right)} + \sum_{j=1, j \neq j'}^{r}\frac{\theta_j^\star}{\left(\theta_j^\star + \frac{\gamma^2}{s_j^2}\right)}\right) \tag{32}$$

$$\leq \frac{\gamma^2}{n}\left(1 + \sum_{j=1, j \neq j'}^{r}\frac{\theta_j^\star}{\left(\theta_j^\star + \frac{\gamma^2}{s_j^2}\right)}\right) \tag{33}$$

$$\leq \frac{\gamma^2}{n}\left(1 + \sum_{j=1, j \neq j'}^{r}\frac{\theta_j^\star}{\left(\frac{\gamma^2}{s_j^2}\right)}\right) \tag{34}$$

$$\leq \frac{\gamma^2}{n}\left(1 + \frac{s_1^2}{\gamma^2}\sum_{j=1, j\neq j'}^{d}\theta_j^\star\right) \quad . \tag{35}$$

Using eq (30) in the above, we have

$$\leq \frac{\gamma^2}{n}\left(1 + \beta\frac{\|\theta^\star\|^2 s_1^2}{\gamma^2}\right) \quad . \tag{36}$$

This completes the proof for eq (9).

**Ridge**: We will now show (11), by characterizing the Excess Risk for the ridge estimator in this regime, showing that it can be upto $r$ times worse. From Lemma I.1, we realize that the Excess Risk expression for the simple ridge estimator $\hat{\theta}(\lambda)$ is given by

$$\forall \lambda > 0, \quad \text{ExcessRisk}\left(\hat{\theta}(\lambda)\right) = \frac{1}{n}\sum_{j=1}^{r}\frac{(\lambda^2\theta_j^\star + \gamma^2 s_j^2)\cdot s_j^2}{(\lambda + s_j^2)^2} \tag{37}$$

$$\geq \underbrace{\frac{\gamma^2}{n}\sum_{j=1}^{r}\frac{s_j^4}{(\lambda + s_j^2)^2}}_{\text{Just the Variance term}} \quad . \tag{38}$$

Inequality (38) above comes from ignoring the bias term (since it is non-negative). Note that the variance term is a *decreasing* function of $\lambda$. And again from Lemma I.1, we get the following expression for $\lambda^\star > 0$ that minimizes the ExcessRisk.

$$\lambda^\star = \gamma^2 \cdot \frac{\sum_{j=1}^{r}\frac{s_j^4}{\left(\lambda^\star + s_j^2\right)^3}}{\sum_{j=1}^{r}\frac{\theta_j^\star s_j^4}{\left(\lambda^\star + s_j^2\right)^3}} \tag{39}$$

$$\leq \gamma^2 \cdot \frac{\sum_{j=1}^{r}\frac{s_j^4}{\left(\lambda^\star + s_j^2\right)^3}}{\frac{\theta_{j'}^\star s_{j'}^4}{\left(\lambda^\star + s_{j'}^2\right)^3}} \qquad \text{(Ignoring positive terms in the denominator above)}$$

$$= \frac{\gamma^2}{\theta_{j'}^\star} \cdot \left(\sum_{j=1}^{r}\frac{s_j^4}{s_{j'}^4}\cdot\frac{\left(\lambda^\star + s_{j'}^2\right)^3}{\left(\lambda^\star + s_j^2\right)^3}\right) \qquad \text{(Rearranging the above)}$$

$$\leq \frac{\gamma^2}{(1-\beta)\|\theta^\star\|^2} \cdot \left(\sum_{j=1}^{r}\frac{s_j^4}{s_{j'}^4}\cdot\frac{\left(\lambda^\star + s_{j'}^2\right)^3}{\left(\lambda^\star + s_j^2\right)^3}\right) \qquad \text{(Using eq (29))}$$

$$= \frac{\gamma^2}{(1-\beta)\|\theta^\star\|^2} \cdot \left(\sum_{j=1}^{j'}\frac{s_j^4}{s_{j'}^4}\cdot\frac{\left(\lambda^\star + s_{j'}^2\right)^3}{\left(\lambda^\star + s_j^2\right)^3} + \sum_{j=j'+1}^{r}\frac{s_j^4}{s_{j'}^4}\cdot\frac{\left(\lambda^\star + s_{j'}^2\right)^3}{\left(\lambda^\star + s_j^2\right)^3}\right)$$

$$\leq \frac{\gamma^2}{(1-\beta)\|\theta^\star\|^2} \cdot \left(\sum_{j=1}^{j'}\frac{s_j^4}{s_{j'}^4} + \sum_{j=j'+1}^{r}\frac{s_j^4}{s_{j'}^4}\cdot\frac{\left(\lambda^\star + s_{j'}^2\right)^3}{\left(\lambda^\star + s_j^2\right)^3}\right) \qquad \text{(Since } s_{j'} \leq s_j \text{ for } j \leq j')$$

$$\leq \frac{\gamma^2}{(1-\beta)\|\theta^\star\|^2} \cdot \left(\sum_{j=1}^{j'}\frac{s_j^4}{s_{j'}^4} + \sum_{j=j'+1}^{r}\frac{s_j^4}{s_{j'}^4}\cdot\frac{s_{j'}^6}{s_j^6}\right) \qquad \text{(Since } \frac{\lambda^\star + s_{j'}^2}{\lambda^\star + s_j^2} \leq \frac{s_{j'}^2}{s_j^2} \text{ for } j > j', \text{ as } \lambda^\star > 0)$$

$$= \frac{\gamma^2}{(1-\beta)\|\theta^\star\|^2} \cdot \left(\underbrace{\sum_{j=1}^{j'}\frac{s_j^4}{s_{j'}^4}}_{\leq r\cdot\left(\frac{s_1}{s_r}\right)^4} + \underbrace{\sum_{j=j'+1}^{r}\frac{s_{j'}^2}{s_j^2}}_{\leq r\cdot\left(\frac{s_1}{s_r}\right)^2}\right)$$

$$\leq \left(\frac{1}{1-\beta}\right) \cdot \frac{\gamma^2}{\|\theta^\star\|^2} \cdot r \left(\left(\frac{s_1}{s_r}\right)^4 + \left(\frac{s_1}{s_r}\right)^2\right)$$

$$\leq \left(\frac{1}{1-\beta}\right) \cdot \frac{\gamma^2}{\|\theta^\star\|^2} \cdot 2r \left(\frac{s_1}{s_r}\right)^4 \quad . \tag{40}$$

Now since the variance term in (38) is a decreasing function of $\lambda$, we can use the upper bound of $\lambda^\star$ from (40) to lower bound the ExcessRisk of optimal ridge as

$$\text{ExcessRisk}\left(\hat{\theta}(\lambda^\star)\right) \geq \frac{\gamma^2}{n} \cdot \sum_{j=1}^{r} \frac{1}{\left(1 + \frac{\lambda^\star}{s_j^2}\right)^2} \qquad \text{(Rewriting eq (38))}$$

$$\geq \frac{r\gamma^2}{n} \cdot \frac{1}{\left(1 + \frac{\lambda^\star}{s_r^2}\right)^2} \qquad \text{(Since } 0 < s_r \leq s_j \text{ for all } j)$$

$$\geq \frac{r\gamma^2}{n} \cdot \frac{1}{\left(1 + \left(\frac{1}{1-\beta}\right) \cdot \frac{2r\gamma^2}{\|\theta^\star\|^2 s_r^2} \cdot \left(\frac{s_1}{s_r}\right)^4\right)^2} \qquad \text{(Using eq (40))}$$

$$\geq \frac{r\gamma^2}{n} \cdot \frac{1}{\left(1 + \frac{1}{99(1-\beta)}\right)^2} \qquad \text{(Using Condition \#2)}$$

$$= \frac{r\gamma^2}{n} \cdot \frac{(99(1-\beta))^2}{(100 - 99\beta)^2} \tag{41}$$

$$= \frac{r\gamma^2}{n} \cdot \underbrace{\left(\frac{99}{100}\right)^2}_{\geq 0.98} \frac{(1-\beta)^2}{(1 - 0.99\beta)^2} \quad . \tag{42}$$

This completes the proof of eq (11).

**1-step SD**: To evaluate 1-step SD's ExcessRisk, we make use of Theorem 5. Observe that since $k = 1$, the ExcessRisk is a simple quadratic in one variable. We will call $\xi^{(1)} \in \mathbb{R}$ as just $\xi$ (similar to eq (6)). And similarly, we will call $M^{(1)}, m^{(1)}$ as just $M, m$, both real-valued. We then have

$$\text{ExcessRisk}\left(\hat{\theta}(\lambda, \xi)\right) = M\xi^2 + 2m\xi + c$$

$$\geq c - \frac{m^2}{M} \quad \forall \xi \in \mathbb{R} , \qquad \text{(By simple quadratic min)}$$

$$= \frac{Mc - m^2}{M} \quad . \tag{43}$$

Note that $M, m, c$ are all functions of $\lambda$. Now we evaluate their expressions. $c$ is simply ExcessRisk of ridge, which we will fetch from Lemma I.1. Evaluate $M$ from Theorem 5:

$$M = \frac{1}{n} \sum_{j=1}^{r} \frac{\left(\frac{\lambda\theta_j^\star}{\rho_j} + \gamma^2\right)}{(1+\rho_j)^2} \cdot C_j(1) \cdot C_j(1)$$

$$= \frac{1}{n} \sum_{j=1}^{r} \frac{\left(\frac{\lambda\theta_j^\star}{\rho_j} + \gamma^2\right)}{(1+\rho_j)^2} \cdot \frac{\rho_j^2}{(1+\rho_j)^2} \qquad \text{(Using } C_j(1) = 1 - \frac{1}{1+\rho_j})$$

$$= \frac{1}{n} \sum_{j=1}^{r} \frac{\left(\lambda\theta_j^\star\rho_j + \gamma^2\rho_j^2\right)}{(1+\rho_j)^4}$$

$$= \frac{1}{n} \left(\sum_{j=1}^{r} \frac{\left(\lambda^2\theta_j^\star s_j^6 + \gamma^2\lambda^2 s_j^4\right)}{(\lambda + s_j^2)^4}\right) \quad . \qquad \text{(Using } \rho_j = \frac{\lambda}{s_j^2})$$

Evaluate $m$ from Theorem 5:

$$m = \frac{1}{n} \sum_{j=1}^{r} \frac{(\lambda \theta_j^\star - \gamma^2)}{(1 + \rho_j)^2} \cdot C_j(1)$$

$$= \frac{1}{n} \sum_{j=1}^{r} \frac{(\lambda \theta_j^\star - \gamma^2)}{(1 + \rho_j)^2} \cdot \frac{\rho_j}{1 + \rho_j} \qquad \text{(Using } C_j(1) = 1 - \tfrac{1}{1+\rho_j})$$

$$= \frac{1}{n} \left( \sum_{j=1}^{r} \frac{(\lambda^2 \theta_j^\star s_j^4 - \gamma^2 \lambda s_j^4)}{(\lambda + s_j^2)^3} \right) \quad . \qquad \text{(Using } \rho_j = \tfrac{\lambda}{s_j^2})$$

We now write the expressions together for comparison, before we use them. That is,

$$n \cdot M = \lambda^2 \sum_{j=1}^{r} \frac{\theta_j^\star s_j^6}{(\lambda + s_j^2)^4} + \lambda^2 \gamma^2 \sum_{j=1}^{r} \frac{s_j^4}{(\lambda + s_j^2)^4} \quad ,$$

$$n \cdot m = \lambda^2 \sum_{j=1}^{r} \frac{\theta_j^\star s_j^4}{(\lambda + s_j^2)^3} - \lambda \gamma^2 \sum_{j=1}^{r} \frac{s_j^4}{(\lambda + s_j^2)^3} \quad ,$$

$$n \cdot c = \lambda^2 \sum_{j=1}^{r} \frac{\theta_j^\star s_j^2}{(\lambda + s_j^2)^2} + \gamma^2 \sum_{j=1}^{r} \frac{s_j^4}{(\lambda + s_j^2)^2} \quad . \qquad \text{(Directly from Lemma I.1)}$$

With all these pieces, for the numerator in eq (43), we have

$$n^2 \cdot (Mc - m^2) = T_1 + T_2 + T_3 \quad ,$$

where we use $T_1, T_2, T_3$ to capture terms of different forms. $T_1$ will capture the $\theta_j^\star$ product terms, $T_2$ will capture the $\gamma^2$ product terms, and $T_3$ will capture the cross terms. Namely,

$$T_1 = \lambda^4 \sum_{j=1}^{r} \sum_{l=1}^{r} \frac{\theta_j^\star \theta_l^\star s_j^2 s_l^2}{(\lambda + s_j^2)^2 (\lambda + s_l^2)^2} \cdot \left( \frac{s_j^4}{(\lambda + s_j^2)^2} + \frac{s_l^4}{(\lambda + s_l^2)^2} - \frac{2 s_j^2 s_l^2}{(\lambda + s_j^2)(\lambda + s_l^2)} \right)$$

$$= \lambda^4 \sum_{j=1}^{r} \sum_{l=1}^{r} \frac{\theta_j^\star \theta_l^\star s_j^2 s_l^2}{(\lambda + s_j^2)^2 (\lambda + s_l^2)^2} \cdot \left( \frac{s_j^2}{(\lambda + s_j^2)} - \frac{s_l^2}{(\lambda + s_l^2)} \right)^2$$

$$\geq 0 \quad . \tag{$\dagger$}$$

$$T_2 = \lambda^2 \gamma^4 \sum_{j=1}^{r} \sum_{l=1}^{r} \frac{s_j^4 s_l^4}{(\lambda + s_j^2)^2 (\lambda + s_l^2)^2} \cdot \left( \frac{1}{(\lambda + s_j^2)^2} + \frac{1}{(\lambda + s_l^2)^2} - \frac{2}{(\lambda + s_j^2)(\lambda + s_l^2)} \right)$$

$$= \lambda^2 \gamma^4 \sum_{j=1}^{r} \sum_{l=1}^{r} \frac{s_j^4 s_l^4}{(\lambda + s_j^2)^2 (\lambda + s_l^2)^2} \cdot \left( \frac{1}{(\lambda + s_j^2)} - \frac{1}{(\lambda + s_l^2)^2} \right)^2$$

$$\geq 0 \quad . \tag{$\dagger\dagger$}$$

$$T_3 = \lambda^2 \gamma^2 \left( \sum_{j=1}^{r} \frac{\theta_j^\star s_j^6}{(\lambda + s_j^2)^4} \right) \left( \sum_{j=1}^{r} \frac{s_j^4}{(\lambda + s_j^2)^2} \right)$$

$$+ \lambda^4 \gamma^2 \left( \sum_{j=1}^{r} \frac{\theta_j^\star s_j^2}{(\lambda + s_j^2)^2} \right) \left( \sum_{j=1}^{r} \frac{s_j^4}{(\lambda + s_j^2)^4} \right)$$

$$+ 2 \lambda^3 \gamma^2 \left( \sum_{j=1}^{r} \frac{\theta_j^\star s_j^4}{(\lambda + s_j^2)^3} \right) \left( \sum_{j=1}^{r} \frac{s_j^4}{(\lambda + s_j^2)^3} \right)$$

$$\geq \lambda^2 \gamma^2 \underbrace{\left( \sum_{j=1}^{r} \frac{\theta_j^\star s_j^6}{(\lambda + s_j^2)^4} \right)}_{Z_1} \underbrace{\left( \sum_{j=1}^{r} \frac{s_j^4}{(\lambda + s_j^2)^2} \right)}_{Y_1} + \lambda^4 \gamma^2 \underbrace{\left( \sum_{j=1}^{r} \frac{\theta_j^\star s_j^2}{(\lambda + s_j^2)^2} \right)}_{Z_2} \underbrace{\left( \sum_{j=1}^{r} \frac{s_j^4}{(\lambda + s_j^2)^4} \right)}_{Y_2} \quad .$$

(Ignoring the $3^{rd}$ term)

We lower bound the terms $Z_1, Y_1$ and $Z_2, Y_2$ in the above lower bound on $T_3$ using simple functional analysis. Consider the fact that the function $f_p(z) := \frac{z^{2p}}{(\lambda+z^2)^p}$ over $z > 0$ is strictly monotonically increasing for any $p \in \mathbb{N}$, since $f'_p(z) = \frac{2p\lambda z^{2p-1}}{(\lambda+z^2)^{p+1}} > 0$ for all $z > 0$. We will use the instantiation $f_4$ for $Z_1$ and $Y_2$, and $f_2$ for $Z_2$ and $Y_1$. For $Z_1$, we get

$$
\begin{aligned}
Z_1 &= \sum_{j=1}^{r} \theta_j^\star \cdot \frac{1}{s_j^2} \cdot \frac{s_j^8}{(\lambda+s_j^2)^4} \\
&\geq \frac{s_r^8}{(\lambda+s_r^2)^4} \sum_{j=1}^{r} \theta_j^\star \cdot \frac{1}{s_j^2} && \text{(Using monotone increasingness of } f_4) \\
&\geq \frac{s_r^8}{(\lambda+s_r^2)^4} \cdot \frac{1}{s_1^2} \underbrace{\sum_{j=1}^{r} \theta_j^\star}_{\geq \theta_{j'}^\star} && \text{(Using monotone decreasingness of } f(z) := {}^1\!/z^2) \\
&\geq \frac{s_r^8}{(\lambda+s_r^2)^4} \cdot \frac{(1-\beta)\|\theta^\star\|^2}{s_1^2} && \text{(Using eq (29))} \\
&\geq (1-\beta)\|\theta^\star\|^2 \cdot \frac{1}{s_1^2} \cdot \frac{s_r^8}{(\lambda+s_r^2)^4} && (44)
\end{aligned}
$$

Following a similar analysis scheme, we get

$$
Z_2 \geq (1-\beta)\|\theta^\star\|^2 \cdot \frac{1}{s_1^2} \cdot \frac{s_r^4}{(\lambda+s_r^2)^2} \quad , \tag{45}
$$

$$
Y_1 \geq r \cdot \frac{s_r^4}{(\lambda+s_r^2)^2} \quad , \tag{46}
$$

$$
Y_2 \geq r \cdot \frac{1}{s_1^4} \cdot \frac{s_r^8}{(\lambda+s_r^2)^4} \quad . \tag{47}
$$

Using eqs (44), (45), (46), (47), we get an overall lower bound on $T_3$. For the overall lower bound on the numerator of eq (43), we combine this with eqs (†), (††) for the terms $T_1$ and $T_2$. We get

$$
n^2 \cdot (Mc - m^2) \geq r\gamma^2 \cdot \frac{(1-\beta)\|\theta^\star\|^2}{s_1^2} \cdot \frac{s_r^{12}}{(\lambda+s_r^2)^6} \left(\lambda^2 + \frac{\lambda^4}{s_1^4}\right) \quad . \tag{48}
$$

And for the denominator in eq (43), we have

$$
\begin{aligned}
n \cdot M &= \lambda^2 \sum_{j=1}^{r} \frac{\theta_j^\star s_j^6}{(\lambda+s_j^2)^4} + \lambda^2\gamma^2 \sum_{j=1}^{r} \frac{s_j^4}{(\lambda+s_j^2)^4} \\
&= \lambda^2 \left(\sum_{j=1}^{r} \frac{\theta_j^\star s_j^6}{(\lambda+s_j^2)^4} + \gamma^2 \sum_{j=1}^{r} \frac{s_j^4}{(\lambda+s_j^2)^4}\right) \quad . \tag{49}
\end{aligned}
$$

Upper Bound $\propto \lambda^2$ from eq (49):

$$
\begin{aligned}
&\leq \lambda^2 \left(\sum_{j=1}^{r} \frac{\theta_j^\star s_j^6}{s_j^8} + \gamma^2 \sum_{j=1}^{r} \frac{s_j^4}{s_j^8}\right) \\
&\leq \lambda^2 \left(\sum_{j=1}^{r} \frac{\theta_j^\star}{s_j^2} + \gamma^2 \sum_{j=1}^{r} \frac{1}{s_j^4}\right) \\
&\leq \lambda^2 \left(\frac{1}{s_r^2} \sum_{j=1}^{r} \theta_j^\star + \frac{\gamma^2}{s_r^4} \sum_{j=1}^{r} 1\right)
\end{aligned}
$$

$$\leq \frac{\lambda^2}{s_r^4} \left( s_r^2 \sum_{j=1}^{r} \theta_j^\star + \gamma^2 \sum_{j=1}^{r} 1 \right) \tag{‡}$$

$$\leq \frac{\lambda^2}{s_r^4} \left( \|\theta^\star\|^2 s_r^2 + r\gamma^2 \right) \leq \frac{\lambda^2}{s_r^4} \left( \|\theta^\star\|^2 s_1^2 + r\gamma^2 \right) \ . \tag{50}$$

Upper Bound $\propto 1/\lambda^2$ from eq (49):

$$\leq \lambda^2 \left( \sum_{j=1}^{r} \frac{\theta_j^\star s_j^6}{\lambda^4} + \gamma^2 \sum_{j=1}^{r} \frac{s_j^4}{\lambda^4} \right)$$

$$= \frac{1}{\lambda^2} \left( \sum_{j=1}^{r} \theta_j^\star s_j^6 + \gamma^2 \sum_{j=1}^{r} s_j^4 \right)$$

$$\leq \frac{1}{\lambda^2} \left( s_1^6 \sum_{j=1}^{r} \theta_j^\star + \gamma^2 s_1^4 \sum_{j=1}^{r} 1 \right) \tag{‡‡}$$

$$\leq \frac{s_1^4}{\lambda^2} \left( \|\theta^\star\|^2 s_1^2 + r\gamma^2 \right) \ . \tag{51}$$

Note that we used $\sum_{j=1}^{r} \theta_j^\star \leq \sum_{j=1}^{d} \theta_j^\star = \|\theta^\star\|^2$ in eqs (‡, ‡‡). Now we put together the numerator lower bound from eq (48), with the denominator upper bound from eq (50) for the first term, and from eq (51) for the second term. We then get

$$n \cdot \frac{Mc - m^2}{M} \geq r\gamma^2 \cdot \frac{(1-\beta)\|\theta^\star\|^2}{s_1^2} \cdot \frac{s_r^{12}}{(\lambda + s_r^2)^6} \left( \frac{s_r^4}{(\|\theta^\star\|^2 s_1^2 + r\gamma^2)} + \frac{\lambda^6}{s_1^8 (\|\theta^\star\|^2 s_1^2 + r\gamma^2)} \right)$$

$$= (1-\beta) \cdot r\gamma^2 \cdot \frac{\|\theta^\star\|^2 s_r^4}{s_1^2 (\|\theta^\star\|^2 s_1^2 + r\gamma^2)} \left( \frac{s_r^{12}}{(\lambda + s_r^2)^6} + \frac{s_r^8}{s_1^8} \cdot \frac{\lambda^6}{(\lambda + s_r^2)^6} \right)$$

$$= (1-\beta) \cdot r\gamma^2 \cdot \frac{s_r^4}{s_1^4} \cdot \underbrace{\frac{1}{\left(1 + \frac{r\gamma^2}{\|\theta^\star\|^2 s_1^2}\right)}}_{Q_1} \underbrace{\left( \frac{s_r^{12}}{(\lambda + s_r^2)^6} + \frac{s_r^8}{s_1^8} \cdot \frac{\lambda^6}{(\lambda + s_r^2)^6} \right)}_{Q_2} \ . \tag{52}$$

Under Condition #2, it holds that $\frac{r\gamma^2}{\|\theta^\star\|^2 s_1^2} \leq \frac{1}{198} \cdot \frac{s_r^6}{s_1^6} \leq \frac{1}{198}$ (since $s_r \leq s_1$). This gives

$$Q_1 \geq \frac{198}{199} \geq 0.99 \ . \tag{53}$$

And now we analyze $Q_2$ using simple calculus. Note that

$$Q_2 = \frac{t_1 + t_2 \lambda^6}{(\lambda + s_r^2)^6} \ , \qquad \text{where } t_1 = s_r^{12}, \quad t_2 = \frac{s_r^8}{s_1^8} \ .$$

Simple calculus shows that this function is minimized at $\bar{\lambda} = \left( t_1/(t_2 s_r^2) \right)^{0.2} = \left( s_1^8 s_r^2 \right)^{0.2}$. And the min value of the function (evaluated at $\bar{\lambda}$) can be used as a lower bound for this, giving

$$Q_2 \geq \frac{t_1}{s_r^2 \left( \bar{\lambda} + s_r^2 \right)^5} = \frac{1}{\left(1 + \left(\frac{s_1}{s_r}\right)^{1.6}\right)^5} \geq \frac{1}{2^5} \left( \frac{s_r}{s_1} \right)^8 \ . \tag{54}$$

Combining eqs (43), (52), (53), (54), we get

$$\forall \lambda > 0, \forall \xi \in \mathbb{R} \quad \text{ExcessRisk}\left( \hat{\theta}(\lambda, \xi) \right) \geq (1-\beta) \cdot \frac{r\gamma^2}{n} \cdot \frac{0.99}{2^5} \cdot \left( \frac{s_r}{s_1} \right)^{12} \ .$$

Using Condition #4 in the above gives the desired eq (10). $\qquad \square$

# F   Proof of Theorem 2

We point the reader to Appendix A for notations used throughout the proof.

*Proof.* Under the condition of $\forall j \in [r], s_j = 1$, we need to analyze the ExcessRisk for both $k$-step SD and ridge. **Denote $Q := \sum_{j=1}^{r} \theta_j^\star$ for simplicity.**

$k$**-step SD**: Since assumption 2.1 is violated, we cannot use Theorem 4 for the $k$-step SD. Instead, we will work with the quadratic expansion of ExcessRisk from Theorem 5. We will rewrite the expressions from Appendix I for the quadratic coefs. Note that $\rho_j = \lambda/s_j^2 = \lambda$ for all $j \in [r]$ becomes independent of $j$ in this case. And so $C_j(i) = 1 - \frac{1}{(1+\lambda)^i}$ for all $j \in [r], i \in [k]$ also becomes independent of $j$. Let $C(i) := 1 - \frac{1}{(1+\lambda)^i}$ denote the coefs $C_j(i)$, since they're now independent of $j$. Let $\omega := [C(1), C(2), \cdots, C(k)] \in \mathbb{R}^k$. Then, we get

$$\forall (i_1, i_2) \in [k] \times [k] \qquad M_{i_1, i_2}^{(k)} = \frac{1}{n} \sum_{j=1}^{r} \frac{\theta_j^\star + \gamma^2}{(1+\lambda)^2} \cdot C(i_1) \cdot C(i_2)$$

$$= \frac{1}{n} \cdot \frac{Q + r\gamma^2}{(1+\lambda)^2} \cdot C(i_1) \cdot C(i_2)$$

$$\implies M^{(k)} = \frac{1}{n} \cdot \frac{Q + r\gamma^2}{(1+\lambda)^2} \cdot \omega \omega^\top \qquad \in \mathbb{R}^{k \times k} \text{ becomes a rank-1 matrix}$$

Similarly, for $m^{(k)}$ we get

$$\forall i_1 \in [k] \qquad m_{i_1}^{(k)} = \frac{1}{n} \sum_{j=1}^{r} \frac{\lambda \theta_j^\star - \gamma^2}{(1+\lambda)^2} \cdot C(i_1)$$

$$\implies m^{(k)} = \frac{1}{n} \cdot \frac{\lambda Q - r\gamma^2}{(1+\lambda)^2} \cdot \omega \qquad \in \mathbb{R}^k$$

And similarly, for $c$ we get

$$c = \frac{1}{n} \cdot \frac{\lambda^2 Q + r\gamma^2}{(1+\lambda)^2} \qquad \in \mathbb{R}$$

Using the above expressions, we rewrite the overall ExcessRisk for any $k$-step SD ($k \geq 1$) as

$$\text{ExcessRisk}\left(\hat{\theta}(\lambda, \xi^{(k)})\right) = \frac{1}{n} \left( \frac{Q + r\gamma^2}{(1+\lambda)^2} \cdot \langle \omega, \bar{\xi}^{(k)} \rangle^2 + 2 \cdot \frac{\lambda Q - r\gamma^2}{(1+\lambda)^2} \cdot \langle \omega, \bar{\xi}^{(k)} \rangle + \frac{\lambda^2 Q + r\gamma^2}{(1+\lambda)^2} \right)$$

We're aiming for a lower bound on the above quantity, so we can first minimize with respect to $\xi^{(k)}$, and then analyze the remaining as a function of $\lambda$. Note that this is a quadratic in the scalar $\langle \omega, \bar{\xi}^{(k)} \rangle$. Since $q(x) := ax^2 + 2bx + c = c - \frac{b^2}{a} + a \left( x + \frac{b}{a} \right)^2 \geq c - \frac{b^2}{a}$, we have

$$\forall k \geq 1, \forall \xi^{(k)} \in \mathbb{R}^k \qquad \text{ExcessRisk}\left(\hat{\theta}(\lambda, \xi^{(k)})\right) \geq \frac{1}{n} \cdot \frac{1}{(1+\lambda)^2} \left( \lambda^2 Q + r\gamma^2 - \frac{(\lambda Q - r\gamma^2)^2}{Q + r\gamma^2} \right)$$

$$= \frac{1}{n} \cdot \frac{1}{(1+\lambda)^2} \left( \frac{\lambda^2 Q r\gamma^2 + Q r\gamma^2 + 2\lambda Q r\gamma^2}{Q + r\gamma^2} \right)$$

$$= \frac{1}{n} \cdot \frac{Q r\gamma^2}{Q + r\gamma^2}$$

$$= \frac{r\gamma^2}{n} \cdot \frac{1}{1 + \frac{r\gamma^2}{Q}}$$

Note this expression is independent of $\lambda$. Hence the above lower bound holds $\forall k \geq 1, \forall \xi^{(k)} \in \mathbb{R}^k, \forall \lambda > 0$. This concludes the proof of eq (12).

**Ridge**: For ridge, we can simply borrow the expression of $c$ from above for its ExcessRisk. That is

$$\text{ExcessRisk}\left(\hat{\theta}(\lambda)\right) = \frac{1}{n} \cdot \frac{\lambda^2 Q + r\gamma^2}{(1+\lambda)^2}$$

Let $\lambda^\star$ denote its minimizer over $\lambda > 0$. Simple calculus gives $\lambda^\star = \frac{r\gamma^2}{Q}$. Plugging this in, we get the same expression

$$\text{ExcessRisk}\left(\hat{\theta}(\lambda^\star)\right) = \frac{r\gamma^2}{n} \cdot \frac{1}{1 + \frac{r\gamma^2}{Q}}$$

This completes the proof of ridge achieving the lower bound in eq (12). $\qquad \square$

## G  Proof of Theorem 3

We point the reader to Appendix A for notations used throughout the proof.

*Proof.* Under the condition of $\forall j \in [r], \theta_j^\star = z > 0$, we need to analyze the ExcessRisk for both $k$-step SD and ridge.

**$k$-step SD**: Again from Theorem 4, any $k$-step SD estimator's ExcessRisk is

$$\forall k \geq 1, \forall \lambda > 0, \forall \xi^{(k)} \in \mathbb{R}^k, \quad \text{ExcessRisk}\left(\hat{\theta}(\lambda, \xi^{(k)})\right) \geq \frac{\gamma^2}{n} \sum_{j=1}^r \frac{\theta_j^\star}{\left(\theta_j^\star + \frac{\gamma^2}{s_j^2}\right)}$$

$$\geq \frac{\gamma^2}{n} \sum_{j=1}^r \frac{1}{\left(1 + \frac{\gamma^2}{z} \cdot \frac{1}{s_j^2}\right)}$$

$$\text{(Since } \forall j \in [r], \theta_j^\star = z)$$

This completes the proof of eq (13).

**Ridge**: Now for the ridge estimator, we will use Lemma I.1. From eq (61), we get an exact expression for $\lambda^\star > 0$ that minimizes the ExcessRisk. Namely

$$\lambda^\star = \frac{\gamma^2}{z}$$

Substituting this in eq (60), we get

$$\text{ExcessRisk}\left(\hat{\theta}(\lambda^\star)\right) = \frac{1}{n} \sum_{j=1}^r \frac{\left((\lambda^\star)^2 \theta_j^\star + \gamma^2 s_j^2\right) \cdot s_j^2}{(\lambda^\star + s_j^2)^2}$$

$$= \frac{\gamma^2}{n} \sum_{j=1}^r \frac{(\lambda^\star + s_j^2) \cdot s_j^2}{(\lambda^\star + s_j^2)^2} \qquad \text{(Since } \lambda^\star \theta_j^\star = \frac{\gamma^2}{z} z = \gamma^2)$$

$$= \frac{\gamma^2}{n} \sum_{j=1}^r \frac{1}{(1 + \frac{\lambda^\star}{s_j^2})}$$

$$= \frac{\gamma^2}{n} \sum_{j=1}^r \frac{1}{(1 + \frac{\gamma^2}{z} \cdot \frac{1}{s_j^2})} \qquad \text{(Substituting } \lambda^\star = \frac{\gamma^2}{z})$$

This completes the proof of ridge achieving the lower bound in eq (13). $\qquad \square$

## H  Proof of Theorem 4

We point the reader to Appendix A for notations used throughout the proof.

*Proof.* The proof of (14) is a simple instantiation of Lemma 4.1. Since the SD estimator is a particular instance of the general $\hat{\theta}(\mathbf{P})$, i.e.

$$\hat{\theta}\left(\lambda, \xi^{(k)}\right) = \hat{\theta}\left(\mathbf{P} \leftarrow \mathbf{P}\left(\lambda, \xi^{(k)}\right) \cdot \mathbf{\Omega}_\lambda^{-1}\right)$$

Eq (14) follows from the lower bound in Lemma 4.1.

For proving the equality being achieved, we will work with the general $k$-step SD estimator, and show that: Under assumption 2.1, $k = r$ steps are sufficient to provide enough freedom to $\xi^{(k)}$ so that the $k$-step SD estimator achieves the lowest possible ExcessRisk.

Using lemma 4.1, note that the condition $\mathbf{P}\left(\lambda, \xi^{(k)}\right) \cdot \Omega_\lambda^{-1} = \mathbf{P}^\star$ is sufficient to ensure that $\hat{\theta}\left(\lambda, \xi^{(k)}\right) = \hat{\theta}(\mathbf{P}^\star)$, which would mean that the $k$-step SD estimator admits the lowest possible ExcessRisk. Since the eigenspaces for both sides of the equation are the same ($\mathbf{U}_d$), we only need to ensure that the eigenvalues match on both sides. That is, we need the following condition (for indices $j \geq r + 1$, there's no condition since lemma 4.1 tells us that any real value of $\tilde{s}_j$ suffices).

$$\forall j \in [r] \qquad \left(1 - \sum_{i=1}^k \bar{\xi}_i^{(k)} \left\{1 - \left(\frac{s_j^2}{\lambda + s_j^2}\right)^i\right\}\right) \cdot \frac{1}{\lambda + s_j^2} = \tilde{s}_j^\star \tag{55}$$

$$\iff \forall j \in [r] \qquad \left(1 - \sum_{i=1}^k \bar{\xi}_i^{(k)} + \sum_{i=1}^k \bar{\xi}_i^{(k)} \left(\frac{s_j^2}{\lambda + s_j^2}\right)^i\right) \cdot \frac{s_j^2}{\lambda + s_j^2} = \frac{\theta_j^\star}{\theta_j^\star + \frac{\gamma^2}{s_j^2}} \tag{56}$$

Let $a_j(\lambda) := \frac{s_j^2}{\lambda + s_j^2}$. Since $\lambda > 0$, $a_j \in (0, 1), \forall j \in [r]$. The above condition can then be written succinctly in matrix form as

$$\underbrace{A^{(k)}(\lambda)}_{\in \mathbb{R}^{r \times k}} \cdot \underbrace{\bar{\xi}^{(k)}}_{\in \mathbb{R}^k} = \underbrace{\alpha(\lambda)}_{\in \mathbb{R}^r} \tag{57}$$

With the following describing the elements of $A^{(k)}(\lambda), \alpha(\lambda)$ as

$$[\alpha(\lambda)]_j := 1 - \left(\lambda + s_j^2\right) \tilde{s}_j^\star \qquad j \in [r]$$
$$\left[A^{(k)}(\lambda)\right]_{j,i} := 1 - (a_j(\lambda))^i \qquad j \in [r], i \in [k]$$

The notation $A^{(k)}(\lambda), \alpha(\lambda)$ explicitly denotes the dependence on $\lambda$.

Now, $A^{(k)}(\lambda)$ being invertible would ensure that $\exists \bar{\xi}^{(k)} \in \mathbb{R}^k$ that makes equation (57) hold true (i.e. the system of equations admits a solution). For that, we need (1) $k = r$ and (2) $A^{(r)}(\lambda)$ being full-rank. The first condition is stated in the lemma. In what follows, we will prove that $\exists \lambda > 0$ such that the second condition is satisfied, i.e. $A^{(r)}(\lambda)$ being full-rank. Decompose $A^{(r)}(\lambda)$ as

$$A^{(r)}(\lambda) = \underbrace{\begin{bmatrix} 1 & 1 & \cdots & 1 \\ 1 & 1 & \cdots & 1 \\ \vdots & \vdots & \ddots & \vdots \\ 1 & 1 & \cdots & 1 \end{bmatrix}_{r \times r}}_{W} - \underbrace{\begin{bmatrix} a_1(\lambda) & a_1(\lambda)^2 & \cdots & a_1(\lambda)^r \\ a_2(\lambda) & a_2(\lambda)^2 & \cdots & a_2(\lambda)^r \\ \vdots & \vdots & \ddots & \vdots \\ a_r(\lambda) & a_r(\lambda)^2 & \cdots & a_r(\lambda)^r \end{bmatrix}_{r \times r}}_{V(\lambda)}$$

Where $W = \mathbf{1}\mathbf{1}^\top$ is the matrix of all ones, and $V(\lambda)$ is akin to the square Vandermonde matrix (only difference being that the standard definition of Vandermonde also has a column of ones).

Using the Matrix Determinant Lemma, we can write

$$\det(A^{(r)}(\lambda)) = \left(1 - \mathbf{1}^\top V(\lambda)^{-1} \mathbf{1}\right) \cdot \det(-V(\lambda))$$

First note that, with the determinant expansion rule based on the first row

$$\det V(\lambda) = \det \begin{bmatrix} 1 & \mathbf{0}^\top \\ \mathbf{1} & V(\lambda) \end{bmatrix}_{(r+1) \times (r+1)}$$

The matrix on the right is exactly the Vandermonde matrix with $a_0 = 0, a_1(\lambda), a_2(\lambda), \cdots a_r(\lambda)$. Using the formula for the det of a standard (with a row of ones) Vandermonde matrix, we get

$$\det V(\lambda) = \prod_{0 \leq i < j \leq r} (a_j(\lambda) - a_i(\lambda)) = \prod_{1 \leq i \leq r} a_i(\lambda) \cdot \prod_{1 \leq i < j \leq r} (a_j(\lambda) - a_i(\lambda))$$

Since $a_j(\lambda) \in (0,1)$ for all $j \in [r]$, and $a_j(\lambda)$'s are all distinct, we conclude that $\det V(\lambda) \neq 0$, i.e. $V(\lambda)$ is full-rank. What remains to show is that the scalar $\left(1 - \mathbf{1}^\top V(\lambda)^{-1}\mathbf{1}\right)$ is non-zero *for some* $\lambda > 0$. With the SVD of $V(\lambda) = CDE^{-1}$ where $C, E$ are orthonormal and $D$ is diagonal with positive entries, one can expand this term as:

$$\mathbf{1}^\top V(\lambda)^{-1}\mathbf{1} = (E^{-1}\mathbf{1})^\top D^{-1}(C^{-1}\mathbf{1}) \implies |\mathbf{1}^\top V^{-1}\mathbf{1}| \geq \frac{1}{d_{max}} \cdot |\langle C^{-1}\mathbf{1}, E^{-1}\mathbf{1}\rangle|$$

Now observe that as $\lambda \to \infty$, $a_j(\lambda) \to 0$ for all $j \in [r]$, which means that $V(\lambda) \to \mathbf{0}_{r \times r}$, which means that (1) $d_{max} \to 0^+$, and (2) $C \leftrightarrow E$ (since $V(\lambda)$ becomes closer to a symmetric matrix, allowing its left and right singular matrices to approach equality to each other). That is, $\frac{1}{d_{max}} \to \infty$ and $\langle C^{-1}\mathbf{1}, E^{-1}\mathbf{1}\rangle \to \|\mathbf{1}\|^2 = r$. This would ensure that $|\mathbf{1}^\top V(\lambda)^{-1}\mathbf{1}| > 1$, meaning that the scalar $(1 - \mathbf{1}^\top V(\lambda)^{-1}\mathbf{1})$ would be non-zero. Hence $\exists \lambda > 0$, such that $A^{(r)}(\lambda)$ is full-rank. $\qquad\square$

### H.1 Proof of Lemma 4.1

*Proof.* The estimator $\hat{\theta}(\mathbf{P})$ expands as

$$\hat{\theta}(\mathbf{P}) = \mathbf{U}_d \tilde{S} S_d^2 \mathbf{U}_d^\top \theta^\star + \mathbf{U}_d \tilde{S} S_d \mathbf{V}_d^\top \boldsymbol{\eta} \ .$$

Expanding the Excess Risk shows

$$\text{ExcessRisk}\left(\hat{\theta}(\mathbf{P})\right) = \underbrace{\sum_{j=1}^d \left(\left(\tilde{s}_j \cdot s_j^2 - 1\right)^2 \cdot \theta_j^\star \cdot w_j\right)}_{Bias} + \underbrace{\gamma^2 \sum_{j=1}^d \left(\tilde{s}_j^2 \cdot s_j^2 \cdot w_j\right)}_{Variance} \ ,$$

where $w_j = \frac{s_j^2}{n}$ for all $j \in [d]$. This is a simple quadratic expression in $\tilde{s}$. Completing the squares gives the desired optimal values of $\tilde{s}_j^\star, j \in [d]$. $\qquad\square$

## I  Quadratic ExcessRisk: detailed version of Theorem 5 and a proof

We point the reader to Appendix A for notations used throughout the proof.

**Theorem 7** (Formal version of Theorem 5). *The Excess Risk is quadratic in $\bar{\xi}^{(k)} \in \mathbb{R}^k$. Namely*

$$\text{ExcessRisk}\left(\hat{\theta}(\lambda, \xi^{(k)})\right) = \left(\bar{\xi}^{(k)}\right)^\top \underbrace{M^{(k)}}_{\in \mathbb{R}^{k \times k}} \left(\bar{\xi}^{(k)}\right) + 2\left(\bar{\xi}^{(k)}\right)^\top \underbrace{m^{(k)}}_{\in \mathbb{R}^k} + c \tag{58}$$

*where the below holds, for indices $(i_1, i_2)$ in $[k] \times [k]$,*

$$M_{i_1,i_2}^{(k)} = \underbrace{\frac{\lambda}{n}\sum_{j=1}^r \frac{\theta_j^\star}{\rho_j(1+\rho_j)^2} \cdot C_j(i_1) \cdot C_j(i_2)}_{B_{i_1,i_2}^{(k)}} + \underbrace{\frac{\gamma^2}{n}\sum_{j=1}^r \frac{1}{(1+\rho_j)^2} \cdot C_j(i_1) \cdot C_j(i_2)}_{V_{i_1,i_2}^{(k)}}$$

$$= \frac{1}{n}\sum_{j=1}^r \frac{\left(\frac{\lambda\theta_j^\star}{\rho_j} + \gamma^2\right)}{(1+\rho_j)^2} \cdot C_j(i_1) \cdot C_j(i_2) \ ,$$

$$m_{i_1}^{(k)} = \underbrace{\frac{\lambda}{n}\sum_{j=1}^r \frac{\theta_j^\star}{(1+\rho_j)^2} \cdot C_j(i_1)}_{b_{i_1}^{(k)}} + \underbrace{\frac{\gamma^2}{n}\sum_{j=1}^r \frac{(-1)}{(1+\rho_j)^2} \cdot C_j(i_1)}_{v_{i_1}^{(k)}}$$

$$= \frac{1}{n}\sum_{j=1}^r \frac{(\lambda\theta_j^\star - \gamma^2)}{(1+\rho_j)^2} \cdot C_j(i_1) \ ,$$

$$c = \underbrace{\frac{\lambda}{n}\sum_{j=1}^r \frac{\theta_j^\star \rho_j}{(1+\rho_j)^2}}_{From\ Bias} + \underbrace{\frac{\gamma^2}{n}\sum_{j=1}^r \frac{1}{(1+\rho_j)^2}}_{From\ Variance}$$

$$= \frac{1}{n} \sum_{j=1}^{r} \frac{\left(\lambda \theta_j^\star \rho_j + \gamma^2\right)}{(1 + \rho_j)^2} \quad .$$

*The $B, b$ and $V, v$ notation is used to indicate the (squared) Bias and Variance terms respectively. Here $\rho_j, j \in [r]$ is used to simplify the notation, and is defined as $\rho_j := \frac{\lambda}{s_j^2}, j \in [r]$.*

*And the coefs $C_j, j \in [r]$ with $i \in [k]$ have the form*

$$C_j(i) := 1 - \frac{1}{(1 + \rho_j)^i} \qquad j \in [r], i \in [k] \quad . \tag{59}$$

*Proof.* We first use the expansion from Lemma D.1. Secondly, expand $\theta^\star = \sum_{j=1}^{d} \langle \theta^\star, \mathbf{u}_j \rangle \mathbf{u}_j$. Using these, we can expand Excess Risk as

$$\text{ExcessRisk}\left(\hat{\theta}(\lambda, \xi^{(k)})\right) = \mathbb{E}_{\boldsymbol{\eta}}\left[\left\|\hat{\theta}(\lambda, \xi^{(k)}) - \theta^\star\right\|_{\hat{\Sigma}_n}^2\right]$$

$$= \sum_{j=1}^{d} \underbrace{\frac{s_j^2}{n}}_{\hat{\Sigma}_n \text{ weighing}} \cdot \theta_j^\star \cdot \left(\frac{s_j^2}{\lambda + s_j^2}\right)^2 \cdot \left(\frac{\lambda}{s_j^2} + \sum_{i=1}^{k} \bar{\xi}_i^{(k)} \cdot \left\{1 - \left(\frac{s_j^2}{\lambda + s_j^2}\right)^i\right\}\right)^2$$

$$+ \sum_{j=1}^{d} \underbrace{\frac{s_j^2}{n}}_{\hat{\Sigma}_n \text{ weighing}} \cdot \gamma^2 \cdot \frac{s_j^2}{(\lambda + s_j^2)^2} \cdot \left(-1 + \sum_{i=1}^{k} \bar{\xi}_i^{(k)} \cdot \left\{1 - \left(\frac{s_j^2}{\lambda + s_j^2}\right)^i\right\}\right)^2$$

Writing down the above expansion and collecting the corresponding quadratic, linear and constant terms' coefficients in the $\bar{\xi}^{(k)}$, give the desired expressions. $\qquad\square$

## I.1 ExcessRisk **for ridge**

Since we will compare the ridge estimator's ExcessRisk to the $k$-step SD estimator, we state the ExcessRisk expression for the ridge estimator (eq (3)) formally here.

**Lemma I.1.** *The ridge estimator $\hat{\theta}(\lambda)$ satisfies*

$$\forall \lambda > 0, \quad \text{ExcessRisk}\left(\hat{\theta}(\lambda)\right) = \frac{1}{n} \sum_{j=1}^{r} \frac{\left(\lambda^2 \theta_j^\star + \gamma^2 s_j^2\right) \cdot s_j^2}{(\lambda + s_j^2)^2} \quad . \tag{60}$$

*And the optimal penalty $\lambda^\star > 0$ that minimizes this* ExcessRisk *satisfies*

$$\lambda^\star = \gamma^2 \cdot \frac{\sum_{j=1}^{r} \frac{s_j^4}{\left(\lambda^\star + s_j^2\right)^3}}{\sum_{j=1}^{r} \frac{\theta_j^\star s_j^4}{\left(\lambda^\star + s_j^2\right)^3}} \quad . \tag{61}$$

*Proof.* Eq (60) is a simple instantiation of Theorem 5 in the vacuous case of $k = 0$. Specifically, the quantity $c$ captures exactly what we need. Borrowing its expression from the detailed theorem statement (Appendix I) gives us eq (60). Now, taking the derivative of the expression in eq (60) and setting it to zero, we get the stated expression for $\lambda^\star > 0$. $\qquad\square$

## J Discussion on synthetic experiments

This section follows up Section 5.1 with more details and examples about the synthetic problem. Figure 8 shows four more settings, with $\gamma \in \{0.125, 0.25\}$ and $\theta^\star \in \{\mathbf{u}_1, 1/\sqrt{2}(\mathbf{u}_1 + \mathbf{u}_2)\}$.

Figure 8a validates that repeated steps of SD do provide a reduction in the excess risk, since the *lowest point* of the curve for each $k$ is reducing as $k$ increases. Observe that at $k = r = 4$ steps of SD, the curves in Figure 8 become flat. This is because we stated Theorem 4 with a "$\exists \lambda > 0$" such that $r$-step SD can achieve the lower bound, but in practice it can happen "$\forall \lambda > 0$".

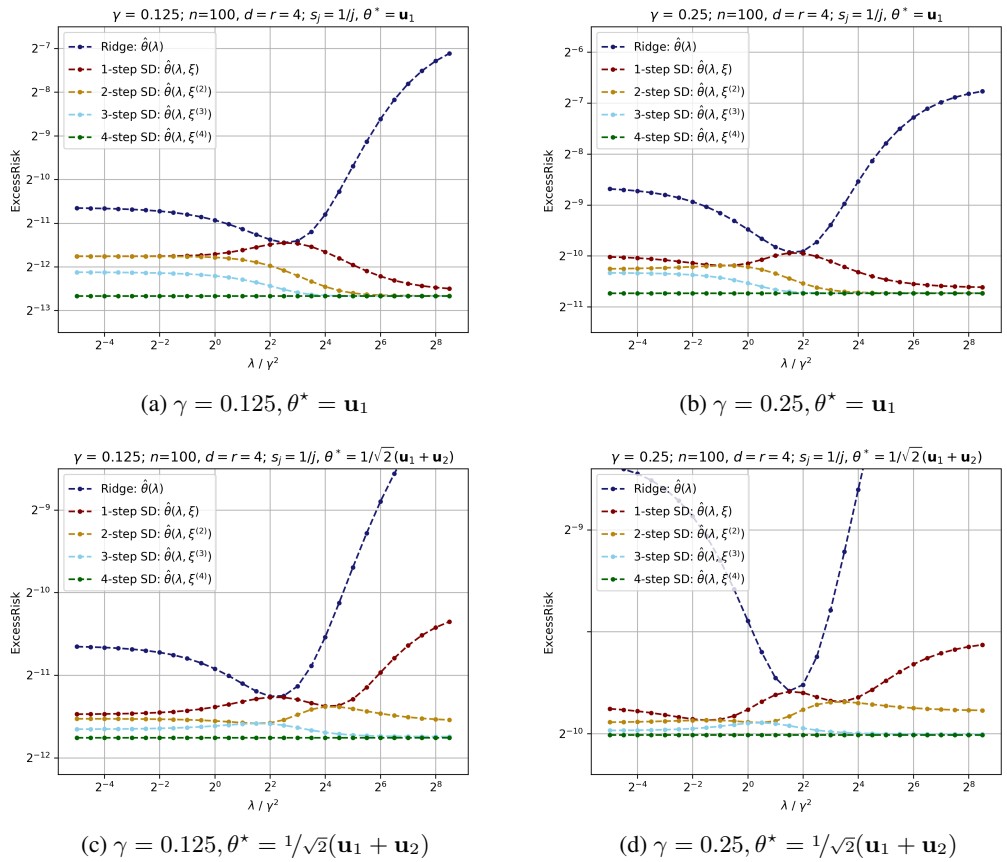

(a) $\gamma = 0.125, \theta^\star = \mathbf{u}_1$

(b) $\gamma = 0.25, \theta^\star = \mathbf{u}_1$

(c) $\gamma = 0.125, \theta^\star = {}^1/\sqrt{2}(\mathbf{u}_1 + \mathbf{u}_2)$

(d) $\gamma = 0.25, \theta^\star = {}^1/\sqrt{2}(\mathbf{u}_1 + \mathbf{u}_2)$

Figure 8: Plot of excess risk of $k$-step SD with optimal $\xi^{(k)}$ for each $\lambda$, for $k \in \{0, 1, 2, 3, 4\}$ on a synthetic problem (Section 5.1).

## K   Discussion on choosing $\xi$ parameters

For the $k$-step SD estimator (eq (8)), for any chosen $\lambda$, Theorem 5 tells us that the ExcessRisk is quadratic in $\bar{\xi}^{(k)} \in \mathbb{R}^k$. To estimate the coefficients of this quadratic from certain chosen evaluations of $\xi^{(k)}$, we need a total of ${}^{k(k+3)}/_2 + 1$ evaluations. This is because the number of unknown coefficients are (i) $k$ for square terms, (ii) ${}^{k(k-1)}/_2$ for cross-square terms, (iii) $k$ for linear terms, and (iv) 1 for the constant term. After estimating the coefficients of the quadratic, we only need to perform a grid search over *one* parameter, which is $\lambda$. We now illustrate this in detail for $k = 1, 2$.

### K.1   Illustration for $k = 1$

For 1-step SD, we know that the ExcessRisk$(\hat{\theta}(\lambda, \xi)) = A\xi^2 + 2B\xi + C$ for unknown $A, B, C$ (that depend on $\lambda$). Training 3 specific 1-step SD estimators, with $\xi = \{-1, 0, 1\}$, and measuring each of those 3 estimators' Risk on the validation set, lets us estimate $A, B, C$. We then know that $\xi^\star = {}^{-B}/_A$, for the chosen value of $\lambda$.

### K.2   Illustration for $k = 2$



Figure 9: Illustrating 2-step SD.

For 2-step SD, the ExcessRisk for a chosen $\lambda$ has the following form

$$\text{ExcessRisk}\big(\hat{\theta}(\lambda, \underbrace{\xi}_{\in \mathbb{R}^2})\big) = A\bar{\xi}_1^2 + B\bar{\xi}_2^2 + 2C\bar{\xi}_1\bar{\xi}_2 + 2D\bar{\xi}_1 + 2E\bar{\xi}_2 + F$$

where the reparametrization is $\bar{\xi}_1 \leftarrow \xi_2(1 - \xi_1)$, and $\bar{\xi}_2 \leftarrow \xi_2\xi_1$ (refer to Appendix C.3 for details on this reparametrization). Let $\text{EVAL}(\xi_2, \xi_1)$ denote the result of measuring this estimator's Risk on the validation dataset. We get the below system of equations.

$$
\begin{aligned}
F &= \text{EVAL}(\xi_2 = 0, \xi_1 = 0) \\
A + 2D + F &= \text{EVAL}(\xi_2 = 1, \xi_1 = 0) \\
A - 2D + F &= \text{EVAL}(\xi_2 = -1, \xi_1 = 0) \\
B + 2E + F &= \text{EVAL}(\xi_2 = 1, \xi_1 = 1) \\
B - 2E + F &= \text{EVAL}(\xi_2 = -1, \xi_1 = 1) \\
\frac{A+B}{4} + \frac{C}{2} + D + E + F &= \text{EVAL}(\xi_2 = 1, \xi_1 = 0.5)
\end{aligned}
$$

The above 6 EVAL operations help us identify all the 6 unknown coefficients. Given $AB - C^2 > 0$ ([15]), we will have the below $\bar{\xi}_1^\star, \bar{\xi}_2^\star$ minimizing the test loss (i.e. giving the optimal 2-step SD coefs for the chosen $\lambda$). And using them, we calculate the actual $\xi_1^\star, \xi_2^\star$ by doing the inverse mapping of the reparametrization.

$$\bar{\xi}_1^\star = \frac{CE - DB}{AB - C^2}, \bar{\xi}_2^\star = \frac{CD - AE}{AB - C^2}$$

$$\xi_1^\star = \frac{\bar{\xi}_2^\star}{(\bar{\xi}_1^\star + \bar{\xi}_2^\star)}, \xi_2^\star = \bar{\xi}_1^\star + \bar{\xi}_2^\star$$

## L   Details on regression experiments

We note that all experiments run on a single CPU within 60 seconds (wall-clock time). We utilize sklearn's implementation of the RIDGE.

**Methodology**. We now explain our methodology in detail, which is followed for all datasets:

1. First, split the original dataset into three parts for a Train-Validation-Test split. We divide all datasets in a $30 - 30 - 40$ split. Note that 30% of the data suffices for training since we work with small datasets which have $d$ on the order of ten (and total number of samples $n$ on the order of thousands). For datasets that have a temporal notion (e.g. date/timestamps), we do the three-way split sequentially.

2. Perform two standard transformations on all three splits of the data: (i) Remove records with missing values, and (ii) coordinate-wise whitening for all $\mathbf{X}$ features and the $\mathbf{Y}$ target, i.e., subtracting the mean (computed on the train set) and dividing by the standard deviation (also computed on the train set).

3. Select a grid of $\lambda$ values (and ensure that it is large enough so that the optimal $\lambda$ lies in it). The grid has a factor of $\sqrt{10}$ difference between consecutive values (e.g., $\{1, \sqrt{10}, 10, \cdots, 10^4\}$). Then, for all $\lambda$ in the grid, compute:

   - The ridge estimator $\hat{\theta}(\lambda)$ using the train set.

   - The 1-step SD estimator $\hat{\theta}(\lambda, \xi^\star)$ using the train set, where $\xi^\star$ for each $\lambda$ is found using the validation set by the strategy described in Section 5.2.

   - The 2-step SD estimator $\hat{\theta}(\lambda, (\xi^{(2)})^\star)$ using the train set, where $(\xi^{(2)})^\star$ for each $\lambda$ is found using the validation set by the strategy described in Appendix K.2.

   Let $\lambda_0^\star$ denote the optimal penalty for ridge that minimizes the MSE on the validation set. Similarly, let $(\lambda_1^\star, \xi^\star)$ and $\big(\lambda_2^\star, (\xi_1^\star, \xi_2^\star)\big)$ denote the optimal parameters for 1-step, 2-step SD, again chosen via the validation set.

4. Evaluate the MSE on the test set (unseen as yet) for all three computed estimators: $\hat{\theta}(\lambda_0^\star)$, $\hat{\theta}(\lambda_1^\star, \xi^\star)$, $\hat{\theta}\big(\lambda_2^\star, (\xi_1^\star, \xi_2^\star)\big)$.

**L.1 Description of the datasets used**

**UCI Air Quality**. The UCI Air Quality dataset [34] contains records of hourly averaged readings from 5 metal oxide chemical sensors (which serve as covariates), along with ground-truth hourly averaged concentrations of the pollutants from a co-located reference certified analyzer. The recordings are from an Italian city over a one year period in 2004-2005. After removing records with missing entries, this dataset has $n = 6941$ rows and we consider $d = 8$ relevant covariates (5 values of the metal oxide chemical sensor readings + 3 values of Temperature, Relative Humidity, and Absolute Humidity). We explicitly mention the field names (all real-numbered non-negative).

- $X$ covariates' names in the dataset: [PT08.S1(CO), PT08.S2(NMHC), PT08.S3(NOx), PT08.S4(NO2), PT08.S5(O3), T, RH, AH]

- $Y$ target's name in the dataset: NO2(GT)

**UCI Airfoil Self-Noise**. The UCI Airfoil Self-Noise dataset [5] contains data obtained from NASA's aerodynamic and acoustic tests of airfoil blade sections. It contains 5 real-valued covariates, which are relevant physical quantities. The target is also real-valued, which is the sound pressure created (in dB). There are no missing entries. This dataset has $n = 1503$ rows and $d = 5$ covariates. We explicitly mention the field names:

- $X$ covariates' names in the dataset: [frequency, attack-angle, chord-length, free-stream-velocity, suction-side-displacement-thickness]

- $Y$ target's name in the dataset: scaled-sound-pressure

**UCI Appliances Energy Prediction**. The UCI AEP dataset [6] contains energy appliances' data collected at 10 min frequency for about 4.5 months. This dataset has no missing entries, and a total of 19735 instances with 28 covariates. We downsample the dataset to hourly frequency, giving $n = 3290$ rows with $d = 24$ relevant covariates (removing degenerate covariates: 'date', 'lights', 'windspeed', 'visiblity'). The full list of covariates is 24 long, and so we do not list it out here (we already mentioned the 4 we removed from the total of 28).

**L.2 Measuring the alignment between $\theta^\star$ and the eigenbasis directions for the datasets**

The goal here is to compute the alignment between the $\theta^\star$ and the eigenbasis directions $\{\mathbf{u}_j\}_{j=1}^d$ for the three real-world datasets. The eigenspace $\mathbf{U}_d$ is known from the design matrix $\mathbf{X}$, but $\theta^\star$ is unknown for real-world tasks. As a *proxy*, we use the ridge solution $\hat{\theta}_\lambda$ (with a small $\lambda$).

**Methodology of choosing** $\lambda$: We considered using the OLS solution $\hat{\theta}_{\text{OLS}} := (\mathbf{XX}^\top)^{-1}\mathbf{XY}$ as the proxy for $\theta^\star$, but $(\mathbf{XX}^\top)^{-1}$ was numerically unstable for these datasets, so we instead used the ridge solution $\hat{\theta}_\lambda$ with a small $\lambda$. We calculated this $\lambda$ methodically for all datasets as a constant fraction of the sum of squared singular values. Explicitly, we $(i)$ computed the SVD of the design matrix $\mathbf{X}$, and $(ii)$ set $\lambda := C \cdot \sum_{j=1}^d s_j^2$ using the obtained singular values. The value $C := 10^{-5}$ was chosen arbitrarily (and the above trend is stable across other reasonably small values of $C$).

Figure 10 shows the result. The sum of all bars in a single plot is *one*, since $\{\mathbf{u}_j\}_{j=1}^d$ are unit-norm vectors that form an orthogonal basis of $\mathbb{R}^d$. We infer two things. **Firstly**, for multi-step SD to outperform ridge (as is the case for the Air Quality and Airfoil datasets), $\theta^\star$ can be well-aligned with *any* of the $\mathbf{u}_j, j \in [d]$; not necessarily $\mathbf{u}_1$. **Secondly**, this gives insight into why multi-step SD could not outperform ridge on the AEP task (Table 1). Unlike the other two datasets, the $\theta^\star$ for the AEP dataset is not strongly aligned with any of the $\mathbf{u}_j, j \in [d]$. The top component in AEP only explains $\sim 35\%$ of the total $\theta^\star$ norm, whereas that number is close to $\sim 80\%$ for the Air Quality and Airfoil datasets. This indicates a large $\beta$ value for the AEP dataset in Assumption 2.2.

**L.3 Observed quadratic nature of MSE w.r.t. $\xi$ parameters**

In this section, we show that the optimal $\xi$ values given in Table 1, selected via the strategy in section 5.2, are indeed the ones that minimize the validation MSE. By inspecting the minima of each of the curves in Figure 11, we observe that the values of $\xi^\star$ for 1-step SD in Table 1 (found through the strategy described in section 5.2) indeed coincide with the minima producing values in the below curves. Further, Figure 11 empirically demonstrates the quadratic nature of risk (MSE) vs $\xi$ described in Theorem 5.

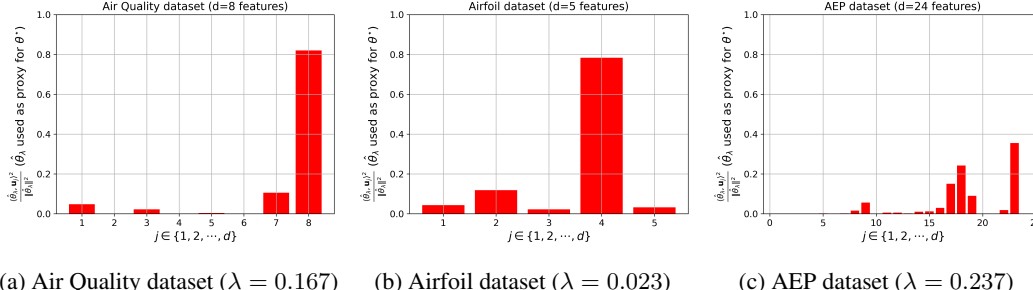

(a) Air Quality dataset ($\lambda = 0.167$)    (b) Airfoil dataset ($\lambda = 0.023$)    (c) AEP dataset ($\lambda = 0.237$)

Figure 10: The alignment of $\theta^\star$ to the eigenbasis directions $\{\mathbf{u}_j\}_{j=1}^d$ for the three datasets used in the experiments. The low-alignment of $\theta^\star$ to any of the eigenbasis directions for the AEP dataset explains why SD provides no gain over ridge in the test set MSE values observed in Table 1. Details of the methodology used to compute the alignment are provided in Appendix L.2.

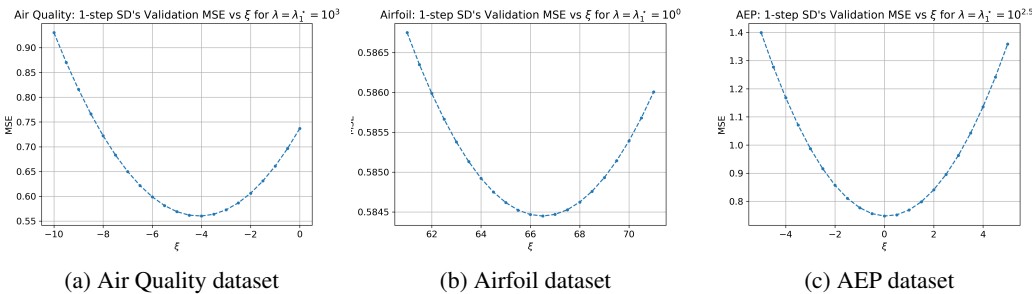

(a) Air Quality dataset    (b) Airfoil dataset    (c) AEP dataset

Figure 11: Observed quadratic nature of MSE (on validation set) of 1-step SD vs $\xi$ for $\lambda = \lambda_1^\star$. This agrees with Theorem 5 and validates the hyperparameter tuning strategy outlined in section 5.2.

