# OpenReview forum: "Understanding the Gains from Repeated Self-Distillation"
_NeurIPS.cc/2024/Conference — NeurIPS 2024 poster_

### Official Review · Reviewer_Uvvg · 2024-07-02

**Soundness:** 3
**Presentation:** 3
**Contribution:** 1
**Rating:** 6
**Confidence:** 4

**Summary:**

This paper theoretically investigates the effect of multiple rounds of self-distillation (SD) in linear regression. Under some conditions on the ground truth model $\theta^{\ast}$ and the data matrix $X$, it is shown that multi-step SD can improve the risk bound by a factor of $r = \text{rank}(X)$. Specifically, this happens when the non-zero singular values of $X$ are all distinct and when $\theta^{\ast}$ is *perfectly* parallel to $u_1$, where $u_1$ is the leading singular vector of $X$. The authors show the necessity of such conditions. The improvement yielded by multi-step SD is demonstrated empirically on some simple regression tasks.

**Strengths:**

**1.** This paper seems to be one of the first ones to theoretically characterize *multi-step* SD.

**2.** The paper is more or less well-written.

**Weaknesses:**

**1.** The condition of $\theta^{\ast}$ being *perfectly* parallel to $u_1$ is too strong and unrealistic in my opinion. This makes the results of this paper less interesting. A better result would be quantifying the gains of multi-step SD assuming the angle between $\theta^{\ast}$ and $u_1$ is bounded by some quantity, say $\beta$ -- this is a more realistic setting. Then, focus on the special case of $\beta$ being small (and so $\beta = 0$ would be a very special case of these general results). Is it possible to do something like this?

**2.** There are no practically relevant insights for multi-step SD in classification problems which are more interesting now.

**3.** Overall, I'm not sure about the relevance of the results in this paper due to the above two points.

**Questions:**

Please see Weakness #1.

**Limitations:**

Not discussed in detail.

---

> ### Author Rebuttal · Authors · 2024-08-07
>
> Thank you for your review. We address your concerns below.
>
> **Concern 1: Relaxing the strong assumption of $\theta^\star$ being perfectly parallel to $\mathbf{u}_1$**
>
> Regarding the condition of $\theta^\star$ being perfectly parallel to $\mathbf{u}_1$, we address this in the shared response, since this was a common concern among reviewers. We agree that it is a strong assumption -- and we presented it as such in the interest of a clean exposition of Theorem 1. At the time of writing, the focus was on showing that there exists *a* regime where $r$-step SD can order-wise outperform $1$-step SD (i.e., not just by a $O(1)$ factor).
>
> But the result does hold true more generally. This is clarified in detail in the shared response via the below two arguments, and a new version of Theorem 1 that incorporates both of these points.
> - First, we note that alignment with *any* one of $\mathbf{u}_j, j \in [r]$ is sufficient for the separation (as noted in lines 203-205 in the manuscript also).
> - Second, based on your valuable recommendation, we have indeed derived a version of Theorem 1 with a relaxed Assumption 2.2, where the closeness of $\theta^\star$ and the singular vector $\mathbf{u}_j$ is controlled with a parameter $\beta$, and setting $\beta = 0$ recovers the special case that we presented.
>
> Through the revised Theorem 1, we argue that the regime of separation is more than just an atypical corner case. Overall, we show that the separation holds whenever the ground-truth $\theta^\star$ is "$\beta$-close" to any of the $\mathbf{u}_j, j \in [r]$; and for small $\beta$. We believe this is a fairly general regime and not just a special case. We hope this provides resolution for your concern.
>
>
> **Concern 2: Insights for self-distillation applied to classification**
>
> We understand that linear regression is a relatively simple task, but we believe that the $\Omega(r)$ order-wise separation between $r$-step SD and $1$-step SD, although in the simple framework of linear regression, is a somewhat surprising result worth sharing with the community.
>
> The de-facto loss for classification problems is the cross-entropy loss, which introduces non-linearity and additional technical challenges for theoretical analysis. But practically, there's a large body of work on empirically analyzing multi-step SD in classification problems. Many researchers have observed the phenomenon of multi-step SD providing performance gains in various settings [1,2,3]. Our work is an attempt to theoretically characterize the gain, and it's surprising that linear regression itself exhibits a non-trivial gain from self-distillation. A detailed analysis for classification with non-linearity falls outside the scope of this work.
>
> ```
> [1] T. Furlanello, Z. Lipton, M. Tschannen, L. Itti, and A. Anandkumar. Born again neural networks. In International Conference on Machine Learning, pages 1607–1616. PMLR, 2018.
> [2] Y. Li, J. Yang, Y. Song, L. Cao, J. Luo, and L.-J. Li. Learning from noisy labels with distillation. In Proceedings of the IEEE international conference on computer vision, pages 1910–1918, 2017.
> [3] Z. Zhang and M. Sabuncu. Self-distillation as instance-specific label smoothing. Advances in Neural Information Processing Systems, 33:2184–2195, 2020.
> ```

---

> > ### Comment · Reviewer_Uvvg · 2024-08-11
> >
> > Thanks for the rebuttal! The revised version of Theorem 1 is what I was looking for. I agree that this is a nice result. So I have updated my score accordingly.

---

> > > ### Author Response · Authors · 2024-08-13
> > > **Thank you**
> > >
> > > Thank you for your review and the valuable suggestion of $\beta$-controlled analysis. We appreciate your reconsideration of this work in light of the rebuttal!

---

### Official Review · Reviewer_2wSn · 2024-07-12

**Soundness:** 3
**Presentation:** 3
**Contribution:** 3
**Rating:** 5
**Confidence:** 3

**Summary:**

The paper analyzes the gains a model can achieve from multi-step self-distillation and presents a solid theory showing that the excess risk does not increase with more self-distillation steps. The synthetic task in the paper effectively proves this analysis.

**Strengths:**

1) The analysis of excess risk for the model trained with multi-step self-distillation is solid.

2) The synthetic task and experiments on the Air Quality, Airfoil, and AEP datasets well support the proposed theory.

3) The paper is well-written and easy to follow.

**Weaknesses:**

1) The study of self-distillation has already been explored in [1,2]. [1] shows that any distillation step can be calculated directly from the initial distillation step. Can the authors compare to [1]? In the experiments of [1], it is shown that accuracy does not always increase with more self-distillation steps. In this paper, the experiments only show results for 2-step self-distillation. How about more steps, like 5 or $\frac{r}{2}$? The datasets used in the experiments do not seem to be widely used. How about results on widely used datasets like CIFAR-10?

2) Assumption 2 is not guaranteed in real tasks. As shown in Figure 2(c), the proposed analysis heavily relies on Assumption 2.2. However, Assumption 2.2 does not seem to be satisfied in real tasks, which weakens the proposed analysis.

3) The experiments are not sufficient to support the proposed theory.

[1] Kenneth Borup and Lars N.Andersen. Even your Teacher Needs Guidance: Ground-Truth Targets Dampen Regularization Imposed by Self-Distillation. In Advances in neural information processing systems, 2020

[2] Rudrajit Das, Sujay Sanghavi. Understanding self-distillation in the presence of label noise. In Proceedings of the 40th International Conference on Machine Learning, 2023.

**Questions:**

1) Please address the weaknesses mentioned.

2) What networks are used in the experiments?

**Limitations:**

Yes, imitations have been discussed.

---

> ### Author Rebuttal · Authors · 2024-08-07
>
> Thank you for your review! We address your concerns and questions below.
>
> **Concern #1.1: Comparison to another relevant paper [1]**
>
> Self-distillation has been explored in [1,2,3], and we do provide a  comparison of our results with [2,3]. Thank you for pointing out the highly relevant reference [1] also! There are definitely similarities, but the main conclusions do differ. Let us elaborate more below.
>
> Indeed [1] also studies multi-step SD similar to us (Fig 1 in both papers are analogous), and they also obtain an analytical form for the k-step SD (Theorem 4.1 in theirs is analogous to Eq. (8) in ours).
>
> The crucial difference is the freedom of the $\xi$ parameters being different at each step of self-distillation. In Lemma 4.2 and Theorem 4.3, [1] assumes the $\xi$ values at each step are equal (denoted $\alpha^{(2)} = \alpha^{(3)} = \cdots = \alpha^{(\tau)}$ in their paper). Consequently, they conclude that subsequent steps of SD progressively sparsify the basis set of obtainable solutions. This means that after a point, running more steps of SD will result in a poorer performing model (similar to [3]), as you pointed out in your review.
>
> Our main result (Theorem 1) is different. It says that subsequent steps of SD strictly provide more freedom, and that the best multi-step SD can outperform the best 1-step SD by an $\Omega(r)$ factor. This separation relies on the freedom of $\xi$s being different at each step, which required careful analysis.
>
> We believe this is a significant novel contribution since an order-wise separation of $\Omega(r)$ (and not just an $O(1)$ difference) is somewhat surprising. But we will surely cite the contributions of [1] as they are highly relevant.
>
> ```
> [1] Kenneth Borup and Lars N.Andersen. Even your Teacher Needs Guidance: Ground-Truth Targets Dampen Regularization Imposed by Self-Distillation. In Advances in neural information processing systems, 2020.
>
> [2] Rudrajit Das and Sujay Sanghavi. Understanding self-distillation in the presence of label noise. In Proceedings of the 40th International Conference on Machine Learning, 2023.
>
> [3] Hossein Mobahi, Mehrdad Farajtabar, and Peter Bartlett. Self-distillation amplifies regularization in hilbert space. In Advances in Neural Information Processing Systems, 2020.
> ```
>
>
> **Concern #1.2: $k$-step SD for $k>2$**
>
> In the regression tasks of Section 5.3, what was challenging for higher $k$-step SD ($k>2$) was numerical instability. In particular, from Theorem 5 we know that the optimal choice of $\xi$ involves computing $M^{-1}$ for the matrix $M \in \mathbb{R}^{k \times k}$, which was unstable to invert for $k > 2$ in the real-world datasets of Section 5.3. There might be more stable ways to approximate the optimal solutions, which will make our approach more practical for higher order SD. We leave this as a future research direction.
>
> As a proof of concept though, we refer you to Figure 2 of the rebuttal pdf, where we run $k$-step SD upto $k=5$ on a synthetic problem. We show that for that specific example, self-distillation beyond $2$-step SD is indeed helpful. In the real-world experiments, we ran only till $2$-step SD because $3$-step SD started becoming numerically unstable for all three of the datasets.
>
> **Concern #2: Assumption 2.2 being too strong**
>
> We agree that Assumption 2.2 as presented is indeed a strong condition, but we do not really need that strong condition to see the gains of multi-step SD. Let us elaborate why:
>
> 1. *Theoretical argument:* In general, we require significantly weaker conditions on $\theta^\star$ for the separation result (Theorem 1). This is clarified in great detail in the shared response (since this was a common concern among reviewers), where we (i) note that alignment with *any* one of $\mathbf{u}_j, j \in [r]$ is sufficient for the separation, and (ii) also present a relaxed version of Theorem 1 with a weaker Assumption 2.2 that does not require *exact* alignment. Please refer to the shared response window for a full discussion.
> 3. *Empirical argument:* The relaxed condition of Assumption 2.2 is fairly reasonable, as shown empirically on the Air Quality and Airfoil regression tasks, where $2$-step SD beats $\{1,0\}$-step SD (especially Air Quality where the gap in performance is significant). We included AEP dataset as a negative example where multi-step SD does not provide gains. Further, Figure 2 of the rebuttal pdf provides insight into why AEP saw no gains. It shows that the $\theta^\star$ is indeed strongly aligned with one of the $\mathbf{u}_j, j \in [d]$ for Air Quality and Airfoil datasets, but *NOT* for the AEP dataset.
>
> We presented Assumption 2.2 as such in the interest of a clean exposition of Theorem 1. At the time of writing, the focus was on showing that there exists *a* regime where $r$-step SD can order-wise outperform $1$-step SD (i.e., not just by a $O(1)$ factor). But the above shows, both theoretically and empirically, that multi-step SD can provide gains in much more general settings.
>
> **Question #2: Models and datasets used**
>
> Since our setting is regression, CIFAR-10 is a less relevant dataset for us. We demonstrate the empirical results on regression datasets taken from the commonly used UCI repository. The experiments presented are for linear estimators of the type $\hat{\theta} \in \mathbb{R}^d$. Non-linear networks were not used to keep the experiments in accordance with the theory.

---

> > ### Comment · Reviewer_2wSn · 2024-08-13
> > **Official Comment by Reviewer 2wSn**
> >
> > Thank you for the reply. Most of my concerns have been addressed.

---

> > > ### Author Response · Authors · 2024-08-13
> > > **Thank you**
> > >
> > > Thank you for your review, and for considering our rebuttal to your concerns. Please let us know if you have any more questions!

---

### Official Review · Reviewer_tBMq · 2024-07-12

**Soundness:** 3
**Presentation:** 3
**Contribution:** 2
**Rating:** 5
**Confidence:** 4

**Summary:**

The paper tries to provide a theoretical analysis of gains from applying self-distiliation repeatedly, in particular by trying to show that it is important to optimally set the imitation paramter for each step rathen than having a fixed Value. The study is conducted on using ridge estimator for linear regressin. The authors provide a theorm on the excess risk under two assumption when multi-step SD is applied. Experimental results on synthetic data and thre real world datasets. The paper also provides insights on how to set the imitation pramter real datasets.

**Strengths:**

Along with the theoretical analysis, the authors provide experimental results to demonstrate the necessity of the two major assumptions they have for the theoretical gain bounds they show. The analysis on importance of properly choosing imitation factor at each step and how it relates to excess risk is insightful. The problem setting, although simplified, but is straightforward to understand and the idea and analysis is well presented.

**Weaknesses:**

The paper does not provide neither discussion nor experiments on how the proposed analysis can be used in more complex setting (e.g. more expressive models). On the other hand, for the experimental results the author do not provide any insight into why on AEP dataset using multi-step SD degrades the results. Could it be due to violation of the assumptions?
In general, the paper provide theoretical analysis for a known technique without providing much insight into how one can use it to improve experimental work, for example it could have been more interesting if the authors could provide more principled techniques for hyperparameter setting for multi-step SD.  It would be more helpful if more experimental analysis on either more datasets or in depth were conducted.

**Questions:**

- Can the authors provide any thoughts on how/challenges of expanding the analysis into more complex family of models?
- How does the Assumption 2.2 can be interpreted as constraints on more complex models, for example for MLPs?
- What could be the reason for drop in performance for AEP?

**Limitations:**

The theoretical analysis is limited in the scope of models and it is not immediately clear how it can be extended into more expressive models. Furthermore, it is not straightforward to incorporate them on large problems by optimally setting hyperparameters.

---

> ### Author Rebuttal · Authors · 2024-08-07
>
> Thank you for your review! We address your questions and concerns below.
>
> **Question 1: Analysis of a more complex model family**
>
> We understand that linear regression is a relatively simple task, but we believe that the $\Omega(r)$ order-wise separation between $r$-step SD and $1$-step SD, although in the simple framework of linear regression, is an interesting and perhaps surprising result.
>
> On extensions, kernel regression is a fairly direct extension in terms of technical tools. The challenge for more expressive models (eg. MLPs) is the introduction of $(i)$ non-linearity and $(ii$) non-convexity, which pose significant additional technical challenges for theoretical analysis. The solution of the ERM problem becomes hard to characterize. Further, there is interplay with optimization since the objective can become non-convex.
>
> Because of these difficulties, researchers generally resort to empirical studies for more complex models. Many past works have observed the phenomenon of multi-step SD providing performance gains in various settings [1,2,3]. Our work is an attempt to theoretically characterize the gain, and it's somewhat surprising that linear regression itself exhibits a non-trivial order-wise gain from self-distillation.
>
> ```
> [1] T. Furlanello, Z. Lipton, M. Tschannen, L. Itti, and A. Anandkumar. Born again neural networks. In International Conference on Machine Learning, pages 1607–1616. PMLR, 2018.
> [2] Y. Li, J. Yang, Y. Song, L. Cao, J. Luo, and L.-J. Li. Learning from noisy labels with distillation. In Proceedings of the IEEE international conference on computer vision, pages 1910–1918, 2017.
> [3] Z. Zhang and M. Sabuncu. Self-distillation as instance-specific label smoothing. Advances in Neural Information Processing Systems, 33:2184–2195, 2020.
> ```
>
> **Question 2: Interpretation of Assumption 2.2 in more complex models**
>
> At a high-level, $\theta^\star$ is akin to the parameterization of the ground-truth generative process (which could be non-linear in general). And $\mathbf{u}_j$ is akin to a basis direction of the observed data (which is computed simply via SVD for the linear case, but the notion of basis could involve non-linearity for more general models). These notions are far from precise though, and it is not clear what the interpretation of their "closeness" (as in Assumption 2.2) would mean in a more complex regime. Characterizing this more explicitly would be a very interesting direction of future work, but falls outside the scope of our current work.
>
> **Question 3: Insight into performance on the AEP dataset**
>
> Note that $\{1, 2\}$-step SD achieve roughly the same test MSE as ridge on the AEP dataset (Table 1 of manuscript), so there is *no drop* in performance, it's just flat. In the attached pdf (Figure 1), we provide an explanation for why this is the case. Unlike the other two datasets, AEP happens to have a $\theta^\star$ which is not strongly aligned with any particular basis direction $\mathbf{u}_j, j \in [d]$. This violates Assumption 2.2, where we require $\theta^\star$ to be well-aligned with one of the $\mathbf{u}_j$ directions for observing large gains from multi-step SD.
> > Please also refer to the shared response for a detailed discussion on Assumption 2.2. In particular, we present a relaxed version of Theorem 1 where $\theta^\star$ needs to be closely (and *not just exactly*) aligned with *any* of the $\mathbf{u}_j$, and not necessarily $\mathbf{u}_1$.
>
> To be more explicit, let us break it into two cases:
> - **Case 1:** From the new version of Theorem 1, we know that closeness of $\theta^\star$ to some $\mathbf{u}_j, j \in [d]$ (which translates to the existence of one high peak in the Figure 1 bar chart) implies large gains from self-distillation.
> - **Case 2:** On the other hand, from Theorem 3 of the manuscript, we know that if $\theta^\star$ is equally aligned to all of $\mathbf{u}_j$s (which translates to a flat distribution in the Figure 1 bar chart), it provably implies no gain from self-distillation.
>
> Since the AEP dataset happened to be closer to the second case, we see no gains from self-distillation. We thank you for this suggestion, and will aim to include this explanation in the revised manuscript.
>
> **Other Concerns: Hyperparameter selection**
>
> The comments also raised a concern about principled hyperparameter selection methods. We would like to highlight that *there is a principled one-shot tuning method we explain in Section 5.2 (and Appendix K) for the $\xi$ parameters*, leveraging the theoretical insights from section 4.4. In particular, from Theorem 5 in section 4.4, we know that the excess risk is quadratic in the $\bar{\xi}$ parameters (one-to-one reparameterization of the $\xi$ parameters). Using the quadratic nature, one can evaluate optimal $\xi$s for any regularization strength $\lambda$, leaving only one parameter ($\lambda$) for the grid search.

---

> > ### Comment · Reviewer_tBMq · 2024-08-13
> >
> > Thank you for the response; I would retain my rating and recommend the clarification on the characteristics of AEP dataset to be include in the main body of the paper.

---

> > > ### Author Response · Authors · 2024-08-13
> > > **Thank you**
> > >
> > > Thank you for your review, especially for the suggestion of analyzing the characteristics of the AEP dataset. This provides valuable insights, and we will include this in the revised manuscript of this work.

---

### Official Review · Reviewer_BMMQ · 2024-07-23

**Soundness:** 4
**Presentation:** 4
**Contribution:** 3
**Rating:** 7
**Confidence:** 3

**Summary:**

This paper explores self-distillation from a theoretical perspective in the context of linear regression. Distillation is when a model is trained simultaneously to predict training labels and the predictions of another model that has already been trained on the data. Self-distillation is when the trained model has the same architecture as the model being trained. There has been recent work empirically show that self-distillation can result in better models and this paper gives a theoretical understanding of multi-step self-distillation in the context of linear regression.

The starting insight of the paper is that repeated self-distillation can be thought of as a pre-conditioner matrix multiplied by the optimal solution to ridge regression. Using this insight, they prove that there is a linear regression problem where self-distillation repeated r times (the rank of the data matrix) can outperform one-step self-distillation and ridge regression by a factor of r. They measure performance in terms of "excess risk" which is the distance to the optimal parameters under a norm weighted by the covariance matrix.

Their example with the factor of r gap requires two assumptions: all the singular values of the data matrix are distinct and the optimal parameters are orthogonal to all but one eigenvector. They show theoretically that both assumptions are necessary for such a gap to exist. They also derive an expression for the excess risk and conjecture that using more than r steps doesn't help the self-distillation process (as long as the self-distillation parameters are chosen optimally?).

They have some experiments where they show that 2-step self-distillation is robust to the ridge regression hyperparameter lambda. Because finding optimal parameters for self-distillation is difficult on real data, they do not show the performance of 3 or more -step self-distillation.

**Strengths:**

* The paper begins with a nice insight into multi-step self-distillation and leverages it to prove several theoretical results.

* The paper shows a very interesting gap in performance between r-step self-distillation and ridge regression and 1-step self-distillation.

* The paper shows that two assumptions in the construction are necessary for the gap.

* The paper shows in practice that 2-step self-distillation works very well.

**Weaknesses:**

Larger weaknesses:

* The paper only shows results in terms of excess risk. I'm not familiar enough with the fixed design setup to determine how useful excess risk is.

* The paper shows that r-step self-distillation can be very powerful but they give no way to practically run it i.e., choose the parameters on real data.

Smaller points to improve:

* I'm confused by the statement of Assumption 2.1. Shouldn't this say that the optimal parameter is orthogonal to every eigenvector *except* the first one?

* More generally, I was confused about the way you stated the assumptions. You referred to two necessary assumptions which I assumed would be Assumption 1 and Assumption 2 but they were actually sub-assumptions in Assumption 2. I would ask that you rework what you call these assumptions and how you refer to them for clarity.

* In your appendix, you referred back to equations in the main body without reproducing them. This made it annoying to check your work e.g., you compared equation 8 and 26 but I kept having to flip between them (until I eventually copied equation 8). Since there's no page limit in the appendix, I would appreciate if you reproduced all equations when you refer to them.

* Your figures are difficult to read when printed out in black and white (and, I assume, for color blind people). Could you please change the marker for each line so they're more distinguishable?

**Questions:**

* You show a gap between 1- and r-step self-distillation. Is there a similar gap between 2-step and r-step self-distillation? Based on your experiments, it looks like 2-step is already very good. Maybe this is a setting where 2-steps basically gives you all the power of r-steps like e.g., load balancing and the "power-of-2 choices".

* You justify using excess risk because it "ensures that the signal-to-noise ratio is uniform in all directions." I don't understand this and I don't see a direct relationship between excess risk and MSE. Could you please tell me why I should care about excess risk? And show me where it comes from theoretically?

**Limitations:**

Yes

---

> ### Author Rebuttal · Authors · 2024-08-07
>
> Thank you for your encouraging review! Let us try to resolve some of your questions and concerns.
>
> **Question 1: Gap between $2$-step SD and $r$-step SD**
>
> The power of two choices in load balancing is related to two choices being the first non-trivial departure from the standard one choice algorithm. Qualitatively (and intuitively), that departure happens perhaps at $1$-step SD in our setting, where the self-distillation is first used. This is why we focused on the separation between $1$-step SD and $r$-step SD in our manuscript, along with the $0$-step (i.e. ridge) versus $r$-step SD. Quantitatively, it is challenging to quantify the gap between $k$-step SD and $r$-step SD for a *general* $k > 1$. Our educated guess is that the gap between each additional step of SD can be upto $O(1)$ (and accumulating the gains over $r$ steps gives an $O(r)$ total gain), but this is not currently proved.
>
> As a proof of concept though, we refer you to Figure 2 of the rebuttal pdf, where we run $k$-step SD upto $k=5$ on a synthetic problem. We show that for that specific example, self-distillation beyond $2$-step SD is indeed helpful. In the real-world experiments, we ran only till $2$-step SD because $3$-step SD started becoming numerically unstable for all three of the datasets.
>
> **Question 2: Justification of using excess risk as the metric**
>
> In linear regression literature, there are two reasons why excess risk in the $\hat{\Sigma}_n$-norm is the preferred metric. The first is that it corresponds to the **test error** on a fresh sample, which is typically the evaluation metric of choice in many settings. The second is that it is **invariant** to the basis of the input $x$. We explain each of these reasons in detail below.
>
> *Test error calculation*
>
> In many cases, we care about the average test error, i.e. the expected error incurred on a fresh sample drawn from the underlying joint distribution $P$. Let's write that mathematically,
> \begin{align}
>     {\rm TotalRisk} (\hat{\theta}) &= \mathbb{E}_{(x,y) \sim P} \left[ \left( \langle \hat{\theta}, x\rangle - y\right)^2 \right] .
> \end{align}
>
> We first break the joint expectation into a conditional expectation on $y \sim P_{Y|X=x}$ followed by $x \sim P_X$.
>
> From Assumption 1 of the paper, we know that $P_{Y|X=x}$ is a distribution with mean $\langle \theta^\star, x \rangle$ and variance $\gamma^2$. So we can write that that $y = \langle \theta^\star, x \rangle + \eta$, where $\eta$ denotes the noise and satisfies $\mathbb{E}[\eta]=0, {\rm Var}[\eta] = \gamma^2$. Using this in the total risk expression, we get
> \begin{align}
>     {\rm TotalRisk} (\hat{\theta}) &= \mathbb{E}_{x \sim P_X} \left[ \langle \hat{\theta} - \theta^\star, x \rangle^2 + \gamma^2 \right] .
> \end{align}
>
> This simplifies to
> \begin{align}
>     {\rm TotalRisk} (\hat{\theta}) &= \left( \hat{\theta} - \theta^\star \right)^\top \mathbb{E}_{x \sim P_X} \left[ xx^\top \right] \left( \hat{\theta} - \theta^\star \right) + \gamma^2 \text{ }.
> \end{align}
>
> Since $P_X$ is the density with point masses at the $n$ training points in the fixed design setting, we get $\mathbb{E}_{x \sim P_X} \left[ xx^\top \right] = \frac{1}{n} \cdot \mathbf{X} \mathbf{X}^\top = \hat{\Sigma}_n$. Using this, we obtain
>
> \begin{align}
>     {\rm TotalRisk} (\hat{\theta}) &= || \hat{\theta} - \theta^\star ||^2_{\hat{\Sigma}_n} + \gamma^2 \text{ }.
> \end{align}
>
> The above calculation shows that excess risk in the $\hat{\Sigma}_n$-norm (i.e. the first term) is the right metric that determines how good an estimator is, above the "*noise floor*" of $\gamma^2$. We remark in Appendix B that some previous works have used the $\ell_2$-norm instead of the more natural $\hat{\Sigma}_n$-norm.
>
> *Invariance*
>
> Another reason is that we prefer a measure of performance that is invariant to what basis we choose for the input $x$. Precisely, we want the performance of an algorithm to be the same whether we apply it to data coming from $(\mathbf{X}, \theta^\star)$ or $(A \mathbf{X}, A^{-\top}\theta^\star)$ for any $d\times d$ invertible matrix $A$. This follows from the fact that one can always apply any $A$ to the input before feeding it to the algorithm, and if the performance changes based on $A$, then it makes it hard to draw a fair comparison. We are assuming there is no numerical instability to focus on the **invariance property**. It happens that the excess risk $|| \hat{\theta} - \theta^\star ||^2_{\hat{\Sigma}_n}$ is invariant to such matrix multiplication (which is the change and/or scaling of the basis). We believe this is also one of the reasons excess risk in the $\hat{\Sigma}_n$-norm is preferred over the $\ell_2$ distance. This is what we meant when we said "*ensures that the signal-to-noise ratio is uniform in all directions*."
>
> **Larger weaknesses**
>
> 1. Please see the above justification for using excess risk as the metric.
> 2. **There is a one-shot tuning method we explain in Section 5.2 (and Appendix K)**, where we delineate a principled hyperparameter selection method for the $\xi$ parameters, leveraging the theoretical insights from section 4.4. In particular, from Theorem 5 in section 4.4, we know that the excess risk is quadratic in the $\bar{\xi}$ parameters (one-to-one reparameterization of the $\xi$ parameters). Using the quadratic nature, one can evaluate optimal $\xi$s for any regularization strength $\lambda$, leaving only one parameter ($\lambda$) for the grid search.
> In practice, what was challenging for higher $k$-step SD ($k>2$) was numerical instability. In particular, from Theorem 5 we know that the optimal choice of $\xi$ would involve computing $M^{-1}$ for the matrix $M \in \mathbb{R}^{k \times k}$, which was unstable to invert for $k > 2$ in the real-world datasets of Section 5.3. There might be more stable ways to approximate the optimal solutions, which will make our approach more practical for higher order SD. We leave this as a future research direction.

---

> > ### Comment · Reviewer_BMMQ · 2024-08-12
> >
> > Thank you for your thorough response! I will retain my score.

---

> > > ### Author Response · Authors · 2024-08-13
> > > **Thank you**
> > >
> > > Thank you for your in-depth review, as well as for considering our rebuttal points.

---

> ### Author Response · Authors · 2024-08-07
> **Addressing the smaller points to improve**
>
> **Smaller points to improve**
>
> Thank you for these comments. We address them below.
>
> 1. **Assumption 2.2 as stated actually *implies* that $\theta^\star$ is orthogonal to every eigenvector except the first one.** We state Assumption 2.2 as $\measuredangle (\theta^\star, \mathbf{u}_1) = 0$, which means that $\theta^\star$ is perfectly parallel to $\mathbf{u}_1$. Since $\mathbf{u}_j, j \in [d]$ are all eigenvectors of $\mathbf{X} \mathbf{X}^T$, they form an *orthonormal basis* of $\mathbb{R}^d$ (also stated in Appendix A), i.e. $\langle \mathbf{u}_i, \mathbf{u}_j \rangle = 0$ for $i, j \in [d], i \neq j$. So indeed, the assumption as stated implies that $\langle \theta^\star, \mathbf{u}_j \rangle = 0$ for $j \in \{2, 3, \cdots, d\}$. There is a separate discussion on whether this condition is too strong, which we address in the shared (global) response window.
> 2. Thank you for catching this! We agree that a clearer way of stating this would explicitly call out the sub-assumptions 2.1 and 2.2. We will work this out in the final draft.
> 3. Totally agreed, we will modify the appendix to have relevant equations handy.
> 4. Also a valid point. We will rework this to a friendlier plotting style (for example, plotting with different marker styles as opposed to different colors).

---

### Author Rebuttal · Authors · 2024-08-07

We thank all the reviewers for their feedback and valuable comments. Multiple reviewers (Uvvg, 2wSn) have raised **concerns about Assumption 2.2 being too strong**, which we address in this shared response. We want to emphasize two points.

- First, as we point out in lines 203-205 in the manuscript, Theorem 1 holds as is for a larger class of problems where $\theta^\star$ is perfectly parallel to *any* one of $\mathbf{u}_j, j \in [r]$. We stated Assumption 2.2 only for $\mathbf{u}_1$ because we viewed it as a lowerbound result. That is, the focus was on showing that there exists *a* regime where the $\Omega(r)$ separation holds. Now that we realize that the result should be stated more generally, we present the following version (to be added in the revised manuscript). This version is stated for any $\mathbf{u}_j$ instead of just $\mathbf{u}_1$.
- Second, we have a new theorem capturing the dependence on the angle between $\theta^\star$ and the most aligned singular vector. Based on a suggestion from the reviewers, we generalize to the case where the angle $\measuredangle (\theta^\star, \mathbf{u}_j)$ for the most aligned $j\in[r]$ is controlled with a parameter $\beta$, instead of the angle being exactly zero. We present the following version of Theorem 1 with a relaxed Assumption 2.2 (that incorporates $\beta$), and Assumption 2.1 staying the same. This will also be added to the revised manuscript. Note that setting $\beta = 0$ exactly recovers the original claim.

We agree that the condition of $\theta^\star$ being perfectly parallel to $\mathbf{u}_1$ was indeed a strong assumption. As mentioned, the focus was on showing *a* regime of separation. Through the above two points (both incorporated in the new theorem below), we argue that the regime of separation is more than just an atypical corner case. Overall, we show that the separation holds whenever the ground-truth $\theta^\star$ is "$\beta$-close" to any of the $\mathbf{u}_j, j \in [r]$; and for small $\beta$. We believe this is a fairly general regime of problem instances.

---

**Assumption 2'**

2'.1 No two non-zero singular values of $\mathbf{X}$ collide, i.e. $s_1 > s_2 > \cdots > s_r > 0$.

2'.2 For some $\beta \in [0, 1)$, there exists an index $j \in [r]$ such that $\langle \theta^\star, \mathbf{u}_j \rangle^2 \geq (1 - \beta) \cdot || \theta^\star ||^2$.

**Theorem 1'**

Under the fixed design linear regression in Assumption 1, there exists a family of problem instances satisfying Assumption 2' such that for any instance $(\mathbf{X}, \theta^\star, \gamma^2)$ in the family, it holds that
\begin{align}
    \exists \lambda > 0, \exists \xi^{(r)} \in \mathbb{R}^r, \hspace{10pt} &{\rm ExcessRisk} \left( \hat{\theta} (\lambda, \xi^{(r)}) \right) \leq \frac{\gamma^2}{n} \cdot \left( 1 \textcolor{red}{ + \beta \text{ } \frac{ || \theta^\star ||^2 s_1^2}{\gamma^2}} \right) , \newline
    \forall \lambda > 0, \forall \xi \in \mathbb{R}, \hspace{10pt} &{\rm ExcessRisk} \left( \hat{\theta} (\lambda, \xi) \right) \geq  \textcolor{red}{(1 - \beta)} \left( \frac{0.99}{2^{9}} \right)  \frac{r\gamma^2}{n} \text{ } , \text{ and } \newline
    \forall \lambda > 0, \hspace{10pt} &{\rm ExcessRisk} \left(\hat{\theta} (\lambda) \right) \geq \textcolor{red}{\left( \frac{1 - \beta}{1 - 0.99 \beta} \right)^2} \left( 0.98 \right)  \frac{r \gamma^2}{n} \text{ } ,
\end{align}
where $r:={\rm rank}(\mathbf{X})$, $n$ is the number of samples, $\hat{\theta}(\lambda,\xi^{(r)})$ and $\hat{\theta}(\lambda,\xi)$ are the $r$-step and $1$-step SD estimators defined in Eqs. (8) and (4) respectively, and $\hat{\theta}(\lambda)$ is the ridge estimator defined in Eq. (3).

---

In this relaxed version, there is an $\Omega(r)$ separation between $r$-step SD and ${1,0}$-step SD in the small $\beta$ regime. In particular, if $\beta$ is $O\left( \frac{\gamma^2}{ || \theta^\star ||^2 s_1^2} \right)$ (which resembles the inverse signal-to-noise ratio) and $\beta << 1$, then $r$-step SD can significantly outperform $1$-step SD.

The proof of this follows the same structure as the proof of Theorem 1 in the paper, with relevant tracking of the inner products $\langle \theta^\star, \mathbf{u}_l \rangle$ for $l \in [r]$.
- Earlier, we had $\langle \theta^\star, \mathbf{u}_1 \rangle^2 = || \theta^\star ||^2$, which changes to $\exists j\in[r]$, $\langle\theta^\star,\mathbf{u}_j\rangle^2\geq(1-\beta)\cdot||\theta^\star||^2$.
- And we had $\langle\theta^\star,\mathbf{u}_l\rangle=0$ for $l\geq2$, which changes to condition (C).

Here (C) is $\sum_{l=1,l\neq j}^r\langle\theta^\star,\mathbf{u}_l\rangle^2\leq\beta\cdot||\theta^\star||^2$. It is worth noting that the bounds in the theorem are tight only in the small $\beta$ regime. As $\beta$ increases above zero, the upper bound and both lower bounds become more loose.

We again thank the reviewers for the valuable suggestion regarding $\beta$-controlled analysis. We hope the above discussion provides some resolution. We will include this generalized version of the result in the revised manuscript.

**Brief note on the attached pdf**

We also upload a 1-page pdf containing two figures.
- Figure 1 is in response to a question from reviewer tBMq about why self-distillation does not provide improvement over ridge on the AEP dataset (Table 1 of the manuscript). We show that unlike the other two datasets, the $\theta^\star$ for AEP is not well-aligned with any of the bases directions $\mathbf{u}_j, j \in [d]$.
- Figure 2 is in response to a question from reviewers BMMQ and 2wSn about the gain from $k$-step SD for $k > 2$. We run upto $5$ steps of self-distillation on a synthetic problem and show that the relative performance gain of $k$-step SD increases beyond $k = 2$ also.

---

### Decision · Program_Chairs · 2024-09-25

**Decision:**

Accept (poster)

**Comment:**

The authors provide a theoretical analysis of repeated (multi-step) self-distillation. This is a variation of the standard distillation setting where labels from a teacher model are used to augment the standard training objective; but this distillation is iterated for several rounds (i.e., distilled students become teachers for the next round of distillation.) For the simplified setting of ridge regression (along with additional assumptions), the authors show that if r is the rank of the data matrix, repeated self-distillation outperforms standard ridge regression by a factor of r.

The overall result is nice and rests on somewhat interesting connections between repeated self-distillation and matrix pre-conditioning in ridge regression. The authors also do a good job in justifying their additional assumptions. The major limitation (of course) is that distillation is usually done for much more complicated models and for typically classification using the cross-entropy loss, so it is as yet an open question whether any of the theoretical insights carry over to practice.